# Development of a versatile nuclease prime editor with upgraded precision

Xiangyang Li[1,2,9], Guiquan Zhang[2,3,9], Shisheng Huang[2,9], Yao Liu[4], Jin Tang[2], Mingtian Zhong[5], Xin Wang[1], Wenjun Sun[1], Yuan Yao[6,7], Quanjiang Ji [8], Xiaolong Wang [4], Jianghuai Liu [3] ✉, Shiqiang Zhu[2] ✉ & Xingxu Huang [1,2] ✉

The applicability of nuclease-based form of prime editor (PEn) has been hindered by its complexed editing outcomes. A chemical inhibitor against DNA-PK, which mediates the nonhomologous end joining (NHEJ) pathway, was recently shown to promote precise insertions by PEn. Nevertheless, the intrinsic issues of specificity and toxicity for such a chemical approach necessitate development of alternative strategies. Here, we find that co-introduction of PEn and a NHEJ-restraining, 53BP1-inhibitory ubiquitin variant potently drives precise edits via mitigation of unintended edits, framing a high-activity editing platform (uPEn) apparently complementing the canonical PE. Further developments involve exploring the effective configuration of a homologous region-containing pegRNA (HR-pegRNA). Overall, uPEn can empower high-efficiency installation of insertions (38%), deletions (43%) and replacements (52%) in HEK293T cells. When compared with PE3/5max, uPEn demonstrates superior activities for typically refractory base substitutions, and for small-block edits. Collectively, this work establishes a highly efficient PE platform with broad application potential.

Genome editing technologies have shown tremendous potential to advance the basic understandings to genetic diseases, and to drive future gene-based therapies[1,2]. Based on the framework of CRISPR–Cas system, different editing tools in the forms of nucleases, base editors, and prime editors, have been harnessed for correcting pathogenic variants in preclinical models[3]. Among these tools, the more recently developed prime editing (PE) technology presents a significant breakthrough. The PE platform employs a prime editor, composed of a Cas9 nickase (H840A mutant, nCas9)/reverse transcriptase fusion protein, and an engineered prime editing guide RNA (pegRNA) derived from the conventional sgRNA. The pegRNA differs from an sgRNA in its

addition of a 3′-extended sequence, which consists of a primer binding site (PBS) and a reverse transcription template (RTT) encoding the intended edits. Following nCas9-mediated nicking and exposure of a 3′-OH group at the target DNA, the pegRNA (via its PBS and RTT) subsequently directs the reverse transcriptase domain to introduce desired edits first into the nicked strand. The ensuing endogenous repair mechanism can lead to permanent installation of edits. In aggregate, PE enables installation of a wide spectrum of genetic modifications, i.e., base substitutions, small insertions and deletions, while acting without the requirements of donor DNA or double-stranded breaks (DSB)[4]. The initial single-nick-dependent PE platform

[1]Gene Editing Center, School of Life Science and Technology, ShanghaiTech University, 100 Haike Rd., Pudong New Area, Shanghai 201210, China. [2]Zhejiang Lab, Hangzhou, Zhejiang 311121, China. [3]State Key Laboratory of Pharmaceutical Biotechnology, Model Animal Research Center at Medical School of Nanjing University, 210061 Nanjing, China. [4]Key Laboratory of Animal Genetics, Breeding and Reproduction of Shaanxi Province, College of Animal Science and Technology, Northwest A&F University, 712100 Yangling, Shaanxi, China. [5]Institute for Brain Research and Rehabilitation, South China Normal University, Guangzhou 510631, China. [6]ZJU-Hangzhou Global Scientific and Technological Innovation Center, Zhejiang University, Hangzhou 311215, China. [7]College of Chemical and Biological Engineering, Zhejiang University, Hangzhou 310027, China. [8]School of Physical Science and Technology, ShanghaiTech University, 100 Haike Rd., Pudong New Area, Shanghai 201210, China. [9]These authors contributed equally: Xiangyang Li, Guiquan Zhang, Shisheng Huang. ✉e-mail: liujianghuai@nju.edu.cn; zhusq@zhejianglab.com; huangxx@shanghaitech.edu.cn

is referred to as PE2, while the PE3 platform entails the addition of a second-strand-nicking sgRNA to facilitate the installation of reverse-transcribed edits[4]. The later iterations of PE4/5 platforms harness the co-introduction of PE2/3 and a mismatch repair-inhibitory hMLH1dn for enhancement of editing efficiency and purity[5]. Various alternative efforts have also been made to improve the original PE2/3 platforms (see review[6]). However, the current nickase-based PE platforms have presented overall suboptimal and inconsistent efficiencies.

The contexts of DNA damage/repair associated with genome editing could strongly impact the editing outcomes[7]. In this regard, the emergence of a variant form of PE, i.e., PE-nuclease (PEn) was noticeable[8,9]. Such platform integrated Cas9 nuclease-generated DSB with RT-dependent synthesis of a 3′ single-stranded DNA (ssDNA) overhang, and was originally developed for bypassing the need of a second nicking sgRNA used otherwise in canonical PE[8]. The initial applications of PEn resulted in sometimes higher levels of accurate edits than PE3, together with a marked induction of unintended edits from the non-homologous end joining (NHEJ) repair of different editing intermediates[8]. The PEn tools have also been applied with dual pegRNAs harboring complementary RTTs for replacing larger genomic regions with shorter designed sequences[9,10]. Despite the potential utilities in certain application contexts, such original versions of PEn warranted further improvements regarding mitigation of unintended edits. Most recently, under the assumption of manipulating the error-prone NHEJ in favor of other potential precise repair pathways[11–15], Peterka et al. applied a chemical inhibitor of DNA-PK (AZD7648) in conjunction with PEn[16]. This was found to substantially reduce the imprecise edits, leading to corresponding increases of precise insertions by PEn[16]. Nevertheless, the uncertainty regarding the chemical inhibitors' specificity/toxicity and their incompatibility with cell-specific or conditional administrations, might limit the development potential of such a modified PEn approach.

In applications of conventional genome editing, protein- or RNAi-based manipulation of the key repair factors have also been adopted to promote precise repair[13,17]. For some repair factors such as tumor suppressor p53-binding protein 1 (53BP1)[18], the detailed mechanisms underlying its stimulatory effects on NHEJ have been well established. Importantly, elegant 53BP1-targeted protein engineering approaches have been developed and demonstrated to enhance conventional genome editing[19–22].

Here, we explore a similar strategy for improvement of PEn. By combining PEn with a 53BP1-inhibitory ubiquitin variant, we establish a platform (ubiquitin variant-assisted PEn, uPEn) with markedly improved efficiencies over PEn and the canonical PE platforms for installing desirable RT-dependent edits.

## Results

### Development of the uPEn system

The recently developed PEn platform presented an interesting example that combination of PE-characteristic writing activity and the Cas9-generated DSB can yield abundant imprecise end-joining repair products containing the templated edits[8–10]. These previous observations suggest that PEn drives potent acquisition of intermediates containing reverse-transcribed DNA for subsequent DSB resolution. We reasoned that such a notable edit-acquiring ability by PEn (more potently than the canonical PE) have provided the basis for its potential refinements toward a high-efficiency genome editing tool.

To confirm the editing profiles by PEn, we first constructed a PEmax nuclease (hereafter referred to as PEn) by reverting the H840A mutation in PEmax[5] to the WT histidine, and designed pegRNAs (containing a EGFP marker) for 3-bp TAG or 18-bp 6*His insertions at the gene loci of *LSP1*, *RUNX1* and *FANCF* (and additionally for 6*His insertion at *SEC61B*). Plasmids encoding PEn/pegRNA were co-transfected into HEK293T cells. For enrichment of edited cells, flow cytometry sorting based on EGFP fluorescence was carried out 3 days

post-transfection. The editing outcomes were determined by targeted deep sequencing analyses. Although application of PEn generally resulted in moderate levels of accurate edits, the large majority of the unintended edits indeed featured partial duplication of RT template sequences (Supplementary Fig. 1a, see example for insertion at *LSP1* site, with >70% of all reads being "imperfect edits"). The WT Cas9-dependent direct indels were also apparently induced by PEn. Such patterns were consistently observed across all sites/edits examined (Supplementary Fig. 1b, c). Here (and in some other initial figures of the manuscript), the total edits that contained pegRNA-templated modifications (both precise and imprecise) are summed as "All RT-driven edits" for comparisons to the levels of "accurate edits". Together, these results validated the notion that PEn stimulates NHEJ repair products containing the templated edits and the classical indels.

With a similar rationale as it from the latest work by Peterka et al.[16], another DNA-PK inhibitor (NU7441[23]) was applied in conjunction with PEn. Consistent with this earlier report, we found that NU7441 enhanced PEn-dependent precise editing in a manner that positively correlated with its dosage (from 3 to 9 μM), while not affecting the levels of total RT-driven edits or classical indels (Supplementary Fig. 1d). Therefore, chemical inhibitor of NHEJ could selectively improve the purity of RT-dependent edits by PEn, reaching a level of up to ~75% precision rate within RT-driven edits at all 3 sites (Supplementary Fig. 1e). The inhibitor was also applied in PEn editing experiments where the pegRNA-programmed 3′ overhang lacked the typical gap-aligning, extended homology to the PAM-proximal end of the DSB (Supplementary Fig. 1f). In contrast to the results above, with such alignment-relaxed pegRNA designs, the application of NU7441 instead reduced the levels of accurate edits, apparently correlated with its effect on "All RT-driven edits" (Supplementary Fig. 1g). These series of results confirm that inhibition of DNA-PK enhances the editing precision of PEn, potentially via a process driven by the homology between the RT-driven sequence and the adjacent DNA end.

While these initial experiments validated the potential of targeting the error-prone NHEJ for enhancement of PEn, it is conceivable that in a practical sense, genetically encoded modulators (as opposed to inhibitory compounds) may be more suited for the use in combination with the expression constructs of PEn. In this regard, 53BP1 might represent a prominent target, as its role in inhibition of DNA end resection, a key event driving NHEJ over the error-free homologous recombination pathway, has been well established[18]. Indeed, Canny et al. recently reported that engineered ubiquitin variants mimicking the K15-ubiquitinated H2A (H2AK15ub), but devoid of canonical ubiquitin properties, could specifically inhibit the recruitment of 53BP1 to DSB region and lead to marked enhancement of Cas9/donor template-dependent precise genome editing[19]. For the present focus on PEn, although the detailed mechanisms underlying its editing activities awaits to be established, we duly tested the effects of 53BP1-sequestering ubiquitin variants on PEn performances (Fig. 1a).

We prepared expression vectors of seven engineered ubiquitin variants (Ubv), including Ub (wt), A10, A11, C08, G08, H04 and G08 (I44A) (Supplementary Fig. 2a). These designations were in reference to those in the original report[19]. To trace their expression, mCherry-P2A peptide was placed at the N terminus of ubiquitin variants. We designed pegRNAs for 34-bp insertions at the gene loci of *LSP1*, *SEC61B*, *RUNX1*. The ubiquitin variants were respectively co-transfected into HEK293T cells with plasmids for PEn and pegRNAs (the latter containing a EGFP marker). For enrichment of cells transfected with all components, EGFP and mCherry double-positive cells were sorted 3 days post-transfection for analyses of editing products. The results showed that, of the tested ubiquitin variants, only the UbvG08 and its derivative G08 (I44A) consistently promoted the levels of accurate edits by PEn at all three endogenous sites (blue bars in Fig. 1b, Supplementary Fig. 2b, d, f), reaching levels of up to 42% [*LSP1*], 50% [*SEC61B*] and 58% [*RUNX1*] in the G08 (I44A) group. On the other

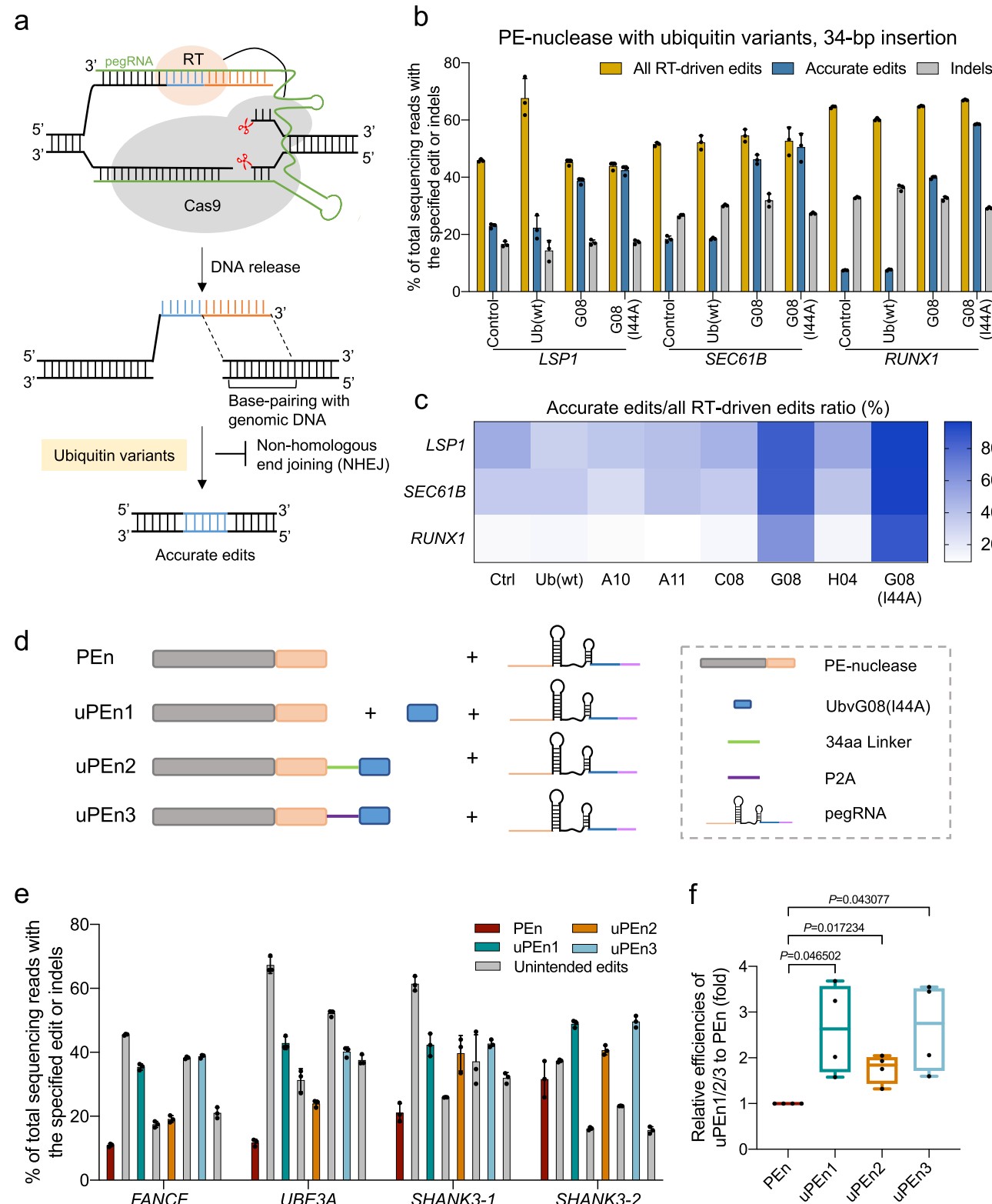

hand, the levels of direct indels (grey bars) or total RT-driven edits (orange bars) were minimally affected by the engineered ubiquitin variants. These analyses indicated that UbvG08 and G08 (I44A) mainly acted via improving the purity of RT-driven edits by PEn. Quantitation of purity within RT-driven edits showed average levels of ~93% in UbvG08 (I44A) group and ~77% in UbvG08 group, in comparison to ~32% in the control group [and similarly in other groups] (Fig. 1c, Supplementary Fig. 2c, e, g). Such effects, particularly those shown by

the UbvG08 (I44A), appeared more robust than the earlier observed effects by DNA-PK inhibitor (see Supplementary Fig. 1d, e). Conversely, the pattern of impurity levels within RT-driven edits among the control, Ub (wt), UbvG08 and G08 (I44A) groups are also demonstrated for presentation purposes (Supplementary Fig. 3a). Relative levels of imprecise RT-driven edits normalized to all sequencing reads in these groups of PEn samples exhibited a very similar trend (Supplementary Fig. 3b). It is worth noting that the G08 (I44A) variant, the most

**Fig. 1 | Design and optimization of uPEn. a** Schematics of PE-nuclease induced DNA repair pathways and the impact by 53BP1-inhibitory ubiquitin variants. Upon Cas9-mediated double-stranded DNA cleavage, the 3′-extension of the pegRNA directs the reverse transcriptase (RT) domain to generate a 3′-overhang structure with the upstream end of the break. Within this overhang, the sequence for insertion is marked in blue, whereas the ensuing segment having the potential to base-pair with the downstream DNA end is marked in orange. Hypothetically, the 53BP-1-inhibory ubiquitin variants could mitigate the non-homologous end-joining repair for such an editing intermediate, therefore potentiating precise insertional edits. The blue segment in the precisely repaired DNA corresponds to that within the 3′ overhang intermediate above. **b** The frequency of all RT-driven edits, accurate edits and indels engaged by PE-nuclease following the co-transfection of Ub (wt), G08 and G08 (I44A) ubiquitin variants at *LSP1*, *SEC61B* and *RUNX1* site in HEK293T cells. All RT-driven edits consist of desired edits and imprecise edits containing the programmed sequence. Accurate edits represent desired edits. Values and error bars reflect the means and standard deviation (s.d.) of three biological replicates. **c** The ratio of accurate edits relative to all RT-driven edits in experimental groups with different ubiquitin variants at *LSP1*, *SEC61B* and *RUNX1* sites. The darker color indicates a higher the percentage of accurate edits.

**d** Schematics of the configuration of three versions of uPEn editing system. PEn, PE-nuclease (Cas9 in grey and RT in light brown). uPEn1, PE-nuclease supplemented with an UbvG08 (I44A) ubiquitin variant that is indicated by the shape in blue. uPEn2, PE-nuclease fused with UbvG08 (I44A) by a 34aa linker (green line). uPEn3, PE-nuclease linked with UbvG08 (I44A) by P2A (purple line). Note that within the illustration for pegRNA, the orange, blue and pink segments represent the spacer, the RTT and the PBS segements, respectively. **e** The editing efficiency of 34-bp fragment insertion at *FANCF*, *UBE3A*, *SHANK3*-1 and *SHANK3*-2 sites in HEK293T cells by PEn, uPEn1, uPEn2, and uPEn3. Note that the "Unintended edits" values include those of the direct indels and the imprecise edits containing the programmed sequences (also used in the ensuing figures). Values and error bars reflect the means and s.d. of three biological replicates. **f** Prime editing efficiencies for insertions by the uPEn systems normalized to the efficiency by PEn at the sites in (**e**). The mean editing frequency induced by PEn for each locus was set to 1, and other samples were normalized correspondingly. The center line shows medians of all data points and the box limits correspond to the upper the lower quartiles, while the whiskers extend to the largest and smallest values. *n* = 4 (sites) for each group. The *P* values (directly marked on the graph) were determined by two-tailed one-sample Student's t-tests. Source data are provided as a Source Data file.

---

effective PEn adjuvant in our hands, was previously named "i53" to denote its optimized competence as a 53BP1 inhibitor[19]. Therefore, our results showed that targeting 53BP1 (via specific Ubv-based inhibitors) under the PEn context would prevent the abundant reverse-transcribed intermediates from error-prone end-joining at the DSB, thus effectively driving precise installation of templated edits. Based on such promising results, we named the combination of PEn and i53 as our first version of ubiquitin variant-assisted PEn (uPEn1).

To further establish an optimal configuration of uPEn, we tested different cloning strategies for introducing the i53 module. The i53 was connected to the C terminus of PEn via a flexible linker [uPEn2] or a cleavable P2A peptide [uPEn3] (Fig. 1d). Such two configurations of i53 expression would lead to PEn-anchored or diffusible inhibitory activities on 53BP1. We compared all three uPEn systems for programming 34-bp insertions at four endogenous sites in HEK293T cells. As i53 selectively affected the purity of RT-driven edits without impacting the levels of classical indels (see above), for simplicity, the levels of all "unintended edits" were summed as one category. This parameter, together with levels of precise edit would be used to inform the performances of uPEn systems (also used frequently in the ensuing figures). Such analyses showed that while all three uPEn systems engaged precise insertions with greater efficiencies compared to PEn, uPEn3 and uPEn1 apparently outperformed uPEn2 (Fig. 1e). This is tightly correlated with the much higher purity of RT-dependent insertional edits by uPEn1 and uPEn3 [-90%] than by uPEn2 [-55%] and PEn [-40%] (Supplementary Fig. 4a–e). With the levels of accurate edits at these 4 sites considered as a whole, the enhancement effects by uPEn1 and uPEn3 over PEn were similar. They were determined to be 2.6- and 2.7-fold (medians), compared to a median enhancement of 1.7-fold by uPEn2 over PEn (Fig. 1f). Such expression format-dependent differences suggested that the free or self-cleaved 53BP1-inhibitory Ubv acted more efficiently. In the ensuing experiments, we further chose to mainly focus on the uPEn3 system due to its more compact configuration. To further rule out potential artifacts shaped by differences in PE protein or corresponding pegRNA levels to account for the changes in editing performances (uPEn *vs* PEn), cells transfected with uPEn3- or PEn-pegRNA [for *UBE3A* site, see Fig. 1e] were harvested for expression analyses. Little difference in the expression of PE protein and the corresponding pegRNA was observed among the three groups (Supplementary Fig. 4f).

Our results have suggested uPEn3 as an effective platform for installing designed insertions. To more informatively demonstrate the performances of uPEn3, we compared the editing efficiencies of PEn/uPEn with those of the conventional Cas9/ssDNA donor strategy and of PEmax (PE2 format, "PE2max") for installing 18-bp insertion at three

endogenous sites in 293 T cells. For the conventional editing strategy, we designed two ssDNA templates. One ssDNA template (ssDNA1) adopted a similar structure as the 3′ extension of pegRNA, which was composed of the sequences corresponding to PBS and RT template (3′ to 5′). In contrast, the other ssDNA template (ssDNA2) was designed as a standard donor ssDNA with respective 35-bp homologous arms flanking the sequence of insertion[24]. As expected from the suboptimal size of the homologous arms in ssDNA1 (20- and 13-bp, respectively) for engagement of HDR, the Cas9/ssDNA1 editing format resulted in the lowest efficiency of precise editing among all groups (Supplementary Fig. 5a–c). The unmodified PEn induced equivalent levels of precise edits as the standard Cas9/ssDNA2 editing format. Notably, the two Cas9/template groups and the PEn group all featured ~70% (medians) unintended edits, consistent with the notion that Cas9/PEn-induced DSB is preferentially repaired by the error-prone NHEJ. Despite the structural differences between Cas9/sgRNA and PEn/pegRNA, the similar indel levels between the Cas9/template groups and the PEn group served as a control to show that these core editing components were likely to exhibit equivalent expression levels in the cells. On the other hand, the application of PE2max led to overall higher levels of precise editing than PEn (respective medians at 30% and 14% for three targets), while it also expectedly featured much lower levels of unintended edits. When the sites were individually considered, PE2max induced higher levels of accurate edits than PEn in two out of three sites. While PE2max displayed a moderate-activity and high-precision profile, the editing patterns by uPEn3 were clearly distinguishable. uPEn not only substantially favored accurate edits over unintended edits in comparison to the PEn and Cas9/ssDNA groups, but also drove the highest levels of precise edits among all groups [median at 55%] (Supplementary Fig. 5a–c). Although the use of uPEn was still associated with higher levels of unintended edits as compared to PE2max, it would be reasonable to perceive that its markedly enhanced efficiencies for installing accurate edits might outweigh such a setback, at least under certain application contexts with the editing efficiency as priorities. These results underscored the advantages by uPEn in its combined capabilities of engaging pegRNA-dependent reverse transcription and of potentiating an error-free DSB repair pathway(s), therefore empowering efficient precise editing.

As an additional control, we evaluated the potential cellular toxicity associated with i53 expression alone or ±PEn components, at the same time point when the editing performances were determined above (72 h post-transfection of HEK293T cells). The expression of i53 alone at different transfected doses did not affect the cell viability. Furthermore, no differences in viability were observed among PE2max, PEn, and uPEn3 groups (Supplementary Fig. 6a). Using Flag-tagged i53,

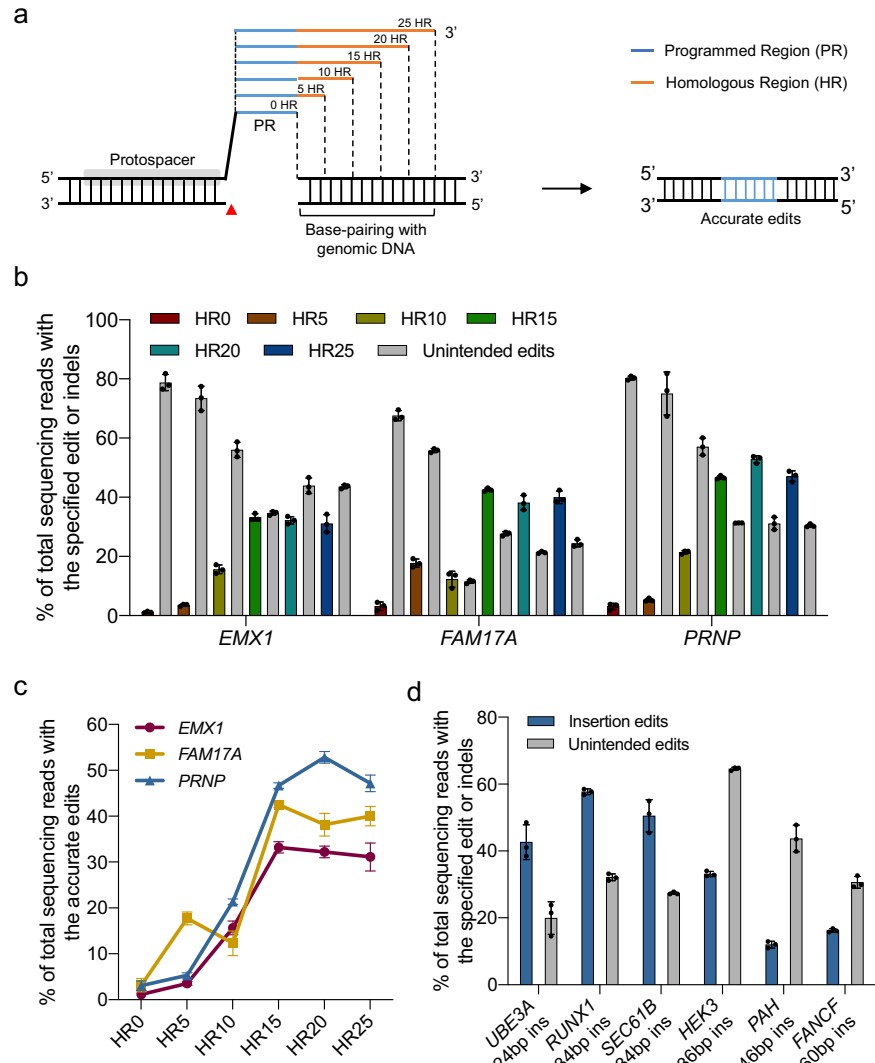

**Fig. 2 | Impacts by the size of the homologous region in pegRNAs on uPEn-dependent editing. a** Schematics of homologous region-containing pegRNAs (HR-pegRNA) with different sizes of the HR (orange). PR, programmed region (blue). Red arrow indicates the position of uPEn-induced DSB. **b** uPEn (in the uPEn3 format, same throughout this figure) editing efficiencies of using pegRNAs with differently sized HR at *EMX1*, *FAM17A* and *PRNP* sites in HEK293T cells. Values and error bars reflect the mean and s.d. of three biological replicates. **c** The pattern of prime editing efficiencies with increasing sizes of HR at three sites in (**b**). Similarly, the values and error bars reflect the mean and s.d. of three biological replicates. **d** Insertional edits of different sequences with uPEn at multiple sites in HEK293T cells. Values and error bars reflect the mean and s.d. of three biological replicates. Source data are provided as a Source Data file.

it was subsequently shown that the expression of i53 remained high 72 h post-transfection, and later dropped substantially on day 5 to 6 (Supplementary Fig. 6b). These results validated the minimal toxicity associated with transient transfections of i53 and of uPEn. The short-lasted expression of i53 following its transient delivery would further limit potential long-term complications.

## Impacts by the size of the homologous region in pegRNAs on uPEn-dependent editing

In regards to uPEn-dependent installation of precise edits, it is conceivable that the homologous region in the RT template segment of the pegRNA would mediate accurate alignment of the broken DNA ends to enable precise edits (Fig. 2a). Therefore, the length of the homologous region may affect the editing outcomes by uPEn. To highlight this feature, we specified the pegRNA used presently as homologous region-containing pegRNA (HR-pegRNA). We next characterized the length of homologous regions that would support high-efficiency uPEn application. For simplicity, we named the inserted sequences in the RT template as programmed region (PR), for

distinguishment from the adjacent HR. Herein, the pegRNAs with different sizes of HR (0, 5, 10, 15, 20, 25 bp) were compared (Fig. 2a). Three different gene loci (*EMX1*, *FAM17A*, and *PRNP*) were targeted for 24-bp insertions. Different pegRNAs were introduced with uPEn3 into HEK293T cells. After sorting (EGFP) of transfected cells, the samples were harvested for NGS analyses (Fig. 2b). The precise editing efficiencies increased with HR sizes from 0 to 15 bp, and plateaued at HR sizes from 15 to 25 bp (Fig. 2c).

Subsequently, we chose 20-bp as a unified HR size for pegRNAs, and finalized the uPEn system for later applications (uPEn3 with HR-pegRNAs). We further characterized the editing efficiency of uPEn at six endogenous sites for variously sized insertions (24- to 60-bp). This HR size-unified uPEn system led to 43%, 58%, 51% and 33% of precise editing at the gene loci of *UBE3A*, *RUNX1*, *SEC61B* and *HEK3*, respectively. Lower editing efficiencies of 12% and 16% were respectively achieved at the gene loci of *PAH* and *FANCF* (Fig. 2d), possibly due to the intrinsic difficulties of installing longer sequence insertion by PE[4].

Given the good efficiencies of precise editing empowered by uPEn, we were interested in investigating the contributing repair

mechanisms. The results above have strongly suggested that the HR segment (with optimal effects reached at -15-bp) within the reverse-transcribed 3′ ssDNA overhang structure is essential for uPEn-associated precise DSB repair. In conventional Cas9/template-mediated genome editing, 53BP1 blockade (or pharmacological NHEJ inhibition) could potentiate precise editing through inducing HDR[13–15,19]. However, since effective HDR processes require operatable templates and at least 30-bp homology between the DSB end and the template[7,25], it is less likely that i53/NU7441-enhanced PEn editing is mediated by the HDR pathways. On the other hand, it is conceivable that such class of NHEJ-targeting enhancement approaches may act upon a more immediate regulatory stage, i.e., to bias the choice between protection and resection of the broken DNA ends toward the latter[7]. Indeed, given the sufficiency of PEn to establish a 3′ ssDNA overhang at the PAM-distal DSB end, the prospective i53/NU7441-stimulated resection of the remaining PAM-proximal DSB end could readily establish the other juxtaposed 3′ ssDNA overhang. Consequently, specific base-pairing between the homologous regions on the pair of overhang structures would yield the key intermediate for the eventual installation of precise edits (Supplementary Fig. 7). Moreover, it is also plausible that under the context of DSB, the associated repair mechanisms would favor the acquisition of RT-driven 3′ overhang sequences, shaping the basis for the more potent activity by uPEn (or NU7441-aided PEn) to install desirable edits than by canonical PE (where the RT-driven 3′-flap structure may be actively rejected[4,5]).

To seek evidence that could possibly report i53-mediated increase of end resection at uPEn-targeted sites, we closely analyzed the patterns of PEn- and uPEn-dependent alleles using the CRISPResso2 program[26]. In the presented example (Supplementary Fig. 8), the editing outcomes for an 18-bp insertion at the *SEC61B* site (with a pegRNA featuring a 20-bp HR) could be explicitly visualized. The overlay of edits along the reference and the desirable product sequences illustrated that the PEn/uPEn-dependent alleles with direct indels or with RT-dependent edits are mutually exclusive (Supplementary Fig. 8a, b, top). The RT-dependent, imprecise edits by PEn harbor indels distal from the guide RNA target. Such distal indels could also be evidently noted in the "indel size distribution graph" in the PEn group (Supplementary Fig. 8a, bottom). Most notably, uPEn was associated with effective mitigation of all PEn-featured distal indels (Supplementary Fig. 8b, bottom). On the other hand, uPEn led to only moderately changed patterns of direct indels in comparison to PEn. The outputs from the "sequence alignment viewer" provided further detailed display of allele distributions following PEn/uPEn (Supplementary Fig. 8c–f). Importantly, it was evident from the display of RT-dependent edits by PEn that the great majority of the distal indels contained the intact 18-bp insertion, in conjunction with variably sized HR fragments (<20-bp) apparently joined directly to the unaltered, downstream blunt end of DSB (Supplementary Fig. 8e). In contrast, with the same low-frequency cutoff, the uPEn group demonstrated much fewer numbers of such directly-joined repair products, with the remaining ones featuring greatly reduced levels (Supplementary Fig. 8f). These particular changes in the patterns of PEn/uPEn-edited alleles are consistent with the model where the uPEn (i53)-promoted resection of the downstream DNA end would create a complementary, downstream overhang to correctly align the RT-dependent 3′-overhang, which would promptly empower a precise repair (see Supplementary Fig. 7).

To examine the rates of DSB end resection from another perspective, we took advantage of our NGS data from the earlier PEn experiments using the specific, HR-free version of pegRNAs [± NU7441] (see Supplementary Fig. 1f, g). We reasoned that while NHEJ inhibition under such condition failed to drive precise repair, the patterns of certain imprecise stitching between the RT-dependent 3′ overhang and the downstream DSB end may be used as genetic scar to independently report the extent of DNA end resection. Therefore,

PEn (±NU7441) with an HR-free pegRNA for targeting again at the *SEC61B* site was subjected to examination. Judged from the "indel size distribution graphs", NU7441 treatment led to moderate changes in the overall patterns of direct and distal indels (Supplementary Fig. 9a, b). Importantly, close inspections of the distal-indel outputs from the "sequence alignment viewer" led to the identification of a deletional allele apparently driven by microhomology between the RTT-encoded overhang and the downstream DSB end (Supplementary Fig. 9c, d, with the microhomology highlighted in blue boxes). Furthermore, NU7441 treatment led to significant increases of such a microhomology-shaped allele ("allele-MH") within total reads and the RT-dependent subset of edits. As a control for allele frequencies, the levels of the most abundant mutant allele ("allele-1", considering the rank in the no-inhibitor group) showed instead substantial decreases (Supplementary Fig. 9e). Since the production of allele-MH reflects the repair mechanism of microhomology-mediated end joining (MMEJ), a process known to be initiated by DSB end resection[27], the higher levels of allele-MH in the NU7441 group provided independent evidence for the increases of downstream DSB end resection upon the combined actions of PEn and NHEJ inhibition. Similar results were obtained from analyses of data at another site [*FANCF*] (Supplementary Fig. 9f–j). Taken together, these allele-type analyses are consistent with a role of i53 (or NU7441)-dependent increase of DSB end resection for enhancement of PEn.

## The uPEn drove various types of prime editing, including insertions, deletions, and replacements with high efficiency

In addition to precise sequence insertion, we hypothesized that uPEn could also efficiently mediate targeted deletions and replacements. The latter types of editing can be conceivably achieved via strategically designing the position of the HR in the RT-templates (Fig. 3a). We tested targeted deletions via uPEn for six different edits at three genomic loci in HEK293T cells. Indeed, uPEn mediated appreciable levels of precise editing for deletion sizes of 24- or 36-bp (76%, 60% and 60%, at sites of *RUNX1*, *HEK3* and *LSP1*, respectively), as well as for larger sizes of 56-, 68-, or 90-bp (24%, 23%, and 18%, at sites of *LSP1*, *HEK3*, and *RUNX1*, respectively) (Fig. 3b). Next, we tested the efficiencies by uPEn for sequence replacements in HEK293T cells. In this series of experiments, uPEn empowered 12-, 26-, 32- [2x] and 46-bp sequence replacements at various loci with 52%, 61%, 46% [average], and 50% efficiencies, respectively (Fig. 3c). When each type of editing at different sites was considered as a group, uPEn drove overall levels (medians) of 38%, 43% and 52% precise editing for these small-block (tens of bp) insertions, deletions and replacements, respectively (Fig. 3d). A summary of all the above three types of editing by the HR size-unified uPEn (see Figs. 2d and 3b, c) yielded a median precise editing efficiency of 49% (Fig. 3d). Such an apparently robust efficiency would be conducive for many applications.

We further tested various types of editing by uPEn in U2OS cells. The levels of precise editing for insertion, replacement and deletion ranged between 12% to 60% (Supplementary Fig. 10a). When the ratios of accurate/all RT-driven edits for individual editing applications were quantitated, the results showed that uPEn led to high-purity desirable products (up to 80%, and at least 60%) within RT-dependent edits in U2OS cells (Supplementary Fig. 10b). Together with the results in HEK293T cells (Figs. 2d and 3b, c), these observations demonstrated the applicability of uPEn in different cellular contexts.

## uPEn mediated more efficient base substitutions at some PE-intractable sites and in PE-resistant cells

One of the most desirable applications of PE is for potentially correcting human pathogenic mutations through somatic tissue/cell editing[3]. Importantly, more than half of currently determined human pathogenic mutations are single-nucleotide variants[4]. With the hitherto encouraging results of uPEn on small-block editing, we next

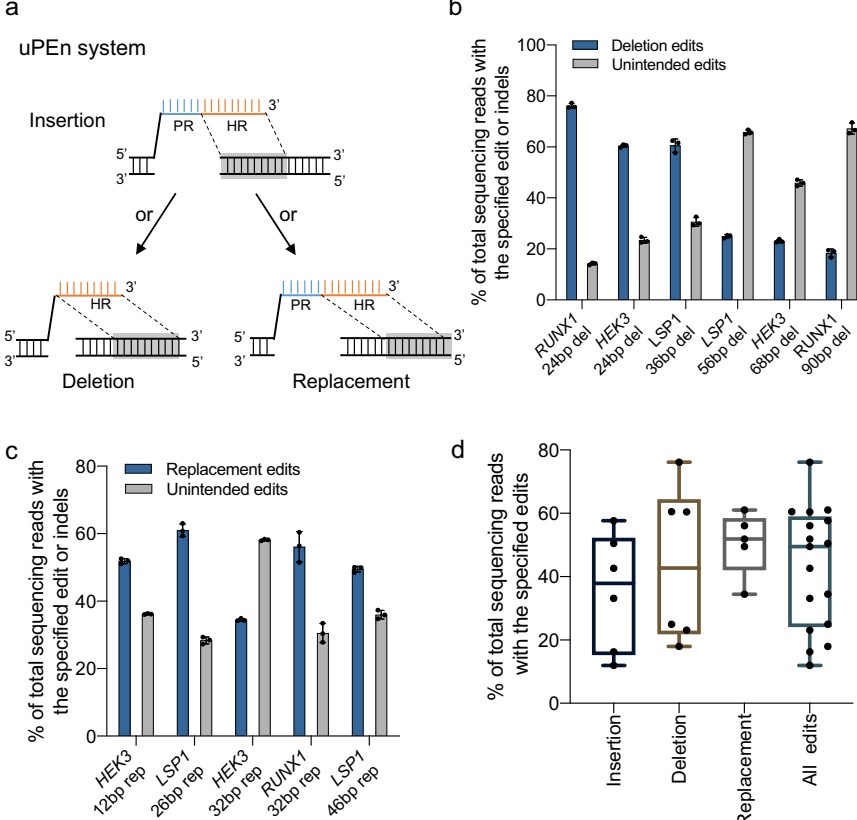

**Fig. 3 | The uPEn mediated more efficient prime editing for insertions, deletions and sequence replacements. a** Schematic diagram illustrating designs and various types of edits (insertions, deletions or replacements) generated by the uPEn system. PR, programmed region (blue); HR, homologous region (orange). Gray fill indicates the base-pairing with the genomic sequence. **b** Deletional edits of different sequences with uPEn (in the uPEn3 format, same throughout this figure) at multiple sites in HEK293T cells. Values and error bars reflect the mean and s.d. of three biological replicates. **c** Sequence replacement type of edits with uPEn at multiple sites in HEK293T cells. Values and error bars reflect the mean and s.d. of three biological replicates. **d** Summary of the insertional ($n = 6$ sites), deletional ($n = 6$ sites) and sequence replacement type of ($n = 5$ sites) editing efficiencies by uPEn in (**b**, **c** and 2**d**). The over efficiencies for all three types of editing ($n = 17$ sites) are also shown. The center line shows medians of all data points and the box limits correspond to the upper the lower quartiles, while the whiskers extend to the largest and smallest values. Source data are provided as a Source Data file.

explored its performances for mediating base substitutions, especially at some PE3max-intractable sites. Thus, PE3max (programmed with pegRNA and nick-sgRNA) and uPEn were respectively designed to target 10 genomic sites (i.e., *ALDOB*, *BCL11A*, *CCR5*, *DNMT1*, *EGFR*, *EMX1*, *KCNA1*, *MECP2*, *RIT1*, *VISTA1*) for various base conversions in HEK293T cells (Fig. 4a). The results showed that uPEn empowered precise base conversions at these 10 targets with a median efficiency of 49%, compared to a corresponding median efficiency of 12% by PE3max (Fig. 4a and Supplementary Fig. 11a). Importantly, among all ten sites, the lowest level of editing by uPEn was 24%, exceeding the 12% median efficiency by PE3max. Such an observation demonstrates a significantly improved base-conversion editing efficiency by uPEn even at sites largely refractory to PE3max. Additionally, when the PE3max activity at each site was set as 1 for normalization, uPEn showed an overall (median) 4.5-fold higher base-conversion activities over PE3max in HEK293T cells (Fig. 4b).

Recently, the PE4/5 platforms were developed on the basis of PE2/3, via co-introducing a mismatch repair-inhibitory MLH1dn protein[5]. These newer platforms exhibited overall enhanced efficiencies in making nucleotide-level edits. Therefore, the PE5max platform would currently represent the state-of-the-art in such pinpointed prime editing. We next compared uPEn and PE5max for installing base substitutions in the MMR-proficient U2OS cells[5]. Here, the U2OS cells were subjected to base conversion at six different target loci with the canonical PE5max or uPEn. Notably, uPEn empowered precise base conversions with a median efficiency of

40%, compared to a corresponding median efficiency of 5% by PE5max (Fig. 4c and Supplementary Fig. 11b). Importantly, among all six sites, the lowest level of editing by uPEn was 27%, exceeding the efficiencies by PE5max at any site. In addition, when the PE5max activity at each site was set as 1 for normalization, uPEn showed an overall (median) 8.2-fold higher base-conversion activities over PE5max in U2OS cells (Fig. 4d).

It is known that PE platforms operate inefficiently in certain cell types, including the commonly used HeLa cells. Therefore, we compared the base conversion efficiencies (at 6 sites) by uPEn and PE5max in HeLa cells. Notably, uPEn empowered precise base conversions with a median level of 14% efficiency, compared to a corresponding level of 2% by PE5max (Supplementary Fig. 12a, b). The uPEn achieved more than 20% (up to 40%) efficiency at 3 out of 6 sites, whereas PE5max led to more than 10% (15% at the maximum) precise edits in only 2 out of 6 sites. While the precise base conversion rates by uPEn remained suboptimal in HeLa cells compared to other cell types (see results in Supplementary Fig. 11a, b), they were significantly higher than those by PE5max in this cell type (with a median improvement of 3.8-fold, Fig. 4e). The nevertheless substandard patterns of accurate/unintended editing ratios by uPEn in HeLa cells (<1 for 5 out of 6 sites, unlike in HEK293T and U2OS cells) point to the existence of other limiting factors for productive applications of PEn in this cell type.

One potential complication associated with the above comparisons between uPEn and PE3/5max is that the delivery of PE3max and PE5max would require greater numbers of plasmids than uPEn.

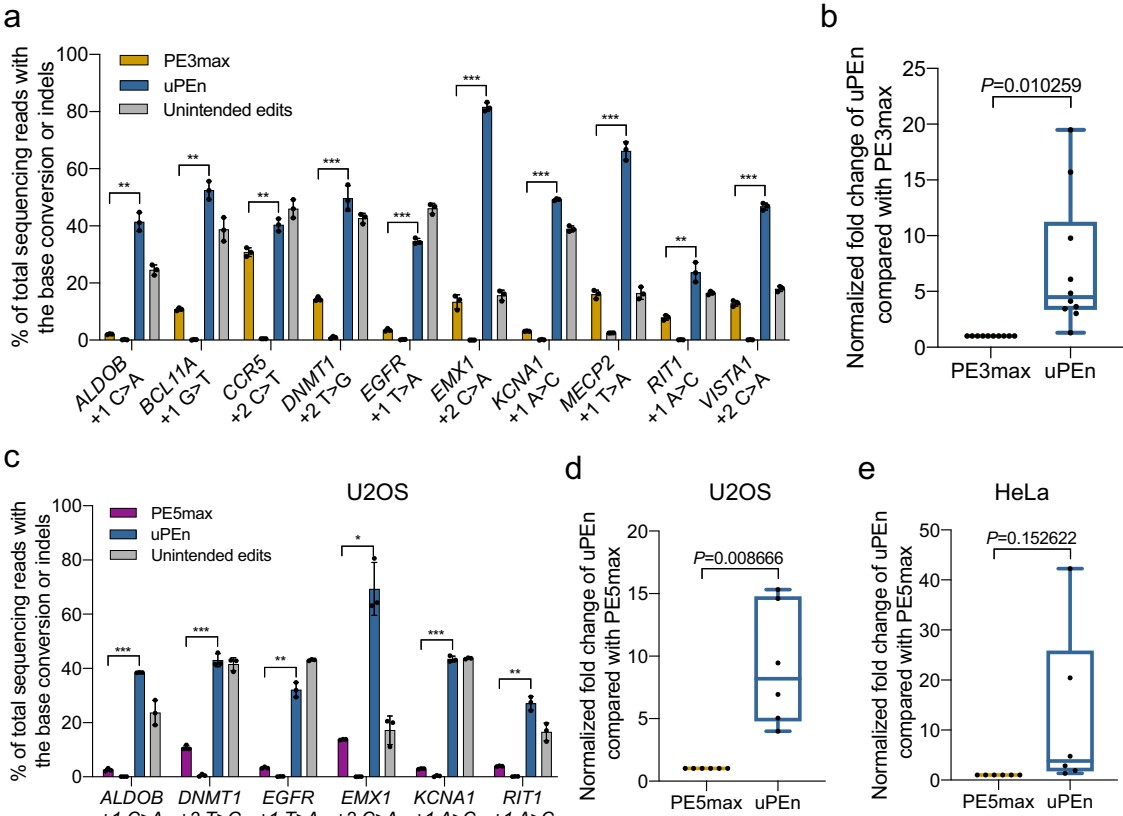

**Fig. 4 | The uPEn mediated efficient base conversions at some largely PE-intractable sites. a** Comparison of base conversion efficiencies and indels induced by PE3max and uPEn (in the uPEn3 format, same throughout this figure) at ten endogenous sites in HEK293T cells. Values and error bars reflect the mean and s.d. of three biological replicates. *P* values were determined (for precise editing efficiencies between PE3max and uPEn at individual sites) by two-tailed Student's t-tests. The asterisks on the graph are used to indicate the ranges (**P* < 0.005, ***P* < 0.0005). The respective *P* values in accordance to their left-to-right order are: 0.002209, 0.001554, 0.003482, 0.000163, 0.000001, 0.000002, 0.000001, 0.000015, 0.001491, 0.000003. **b** Summary of the fold change in uPEn efficiency normalized to PE3max at the same target sites in (**a**). The mean editing frequency induced by PE3max for each locus was set to 1, and other samples were normalized correspondingly. The center line shows medians of all data points and the box limits correspond to the upper the lower quartiles, while the whiskers extend to the largest and smallest values. *n* = 10 (sites) for each group. The *P* value (directly marked on the graph) was determined by a two-tailed one-sample Student's t-test.

**c** Comparison of base conversion efficiencies and indels induced by PE5max and uPEn at six endogenous sites in U2OS cells. Values and error bars reflect the mean and s.d. of three biological replicates. *P* values were determined by two-tailed Student's t-tests. The asterisks on the graph are used to indicate the ranges (**P* < 0.05, ***P* < 0.005, ***P* < 0.0005). The respective *P* values in accordance to their left-to-right order are: 0.000054, 0.000025, 0.00265, 0.010059, 0.000194, 0.003916. **d** Summary of the fold change in uPEn efficiency normalized to PE5max at the same target sites in (**c**). The mean editing frequency induced by PE5max for each locus was set to 1, and other samples were normalized correspondingly. The box plot was generated with the same settings as in (**b**). *n* = 6 (sites) for each group. The *P* value (directly marked on the graph) was determined by a two-tailed one-sample Student's t-test. **e** Summary of base conversion efficiencies of uPEn normalized to PE5max in HeLa cells. The box plot was generated with the same settings as in (**b**). *n* = 6 (sites) for each group. The *P* value (directly marked on the graph) was calculated by a two-tailed one-sample Student's t-test. Source data are provided as a Source Data file.

Therefore, additional control experiments were carried out. For simplification of the transfection step for PE5max, a singular construct was previously established by connecting the PEmax and the MLH1dn modules via P2A[5]. We used the abbreviation of "PE5maxP" (with the "P" denoting P2A) to specify the application of this construct together with pegRNA and nick-sgRNA in transfection. Furthermore, we also established dual-guide RNA-expressing constructs by placing the pegRNA and nick-sgRNA in a same plasmid (respectively under a U6 promoter). Co-transfection of PE5maxP with such a construct would be specified as "PE5maxP-2U6". In this configuration, the PE5max platform could be transfected via 2 plasmids, a format similar to the delivery of uPEn. Next, focusing on three different base-conversional edits previously analyzed by others[5], we compared the effects by PE3max, PE5maxP, PE5maxP-2U6 and uPEn in HEK293T cells (Supplementary Fig. 13). Across these sites, PE5maxP led to only slight increases in efficiencies over PE3max, generally consistent with the previous report[5]. In addition, the combination of guide RNAs in a same plasmid (PE5maxP-2U6) did not result in higher editing levels than those by PE5maxP, excluding the combination of plasmids as a burden

for canonical PE. Importantly, the uPEn drove markedly higher levels of precise edits than PE5maxP-2U6 (and all other groups) in two out of three sites (Supplementary Fig. 13). At the remaining site (within *CXCR4* locus), uPEn enabled an equivalent level of editing in comparison to that by PE5maxP-2U6. Interestingly, we noted that PE3/5max activities were already robust (~60%) at this site. Here, the overall higher activities of uPEn compared to various PEmax platforms or configuration-simplified versions, especially in installing less editable base conversions (see sites of *CDKL5* and *IL2RB*), further corroborated our earlier results in HEK293T, U2OS and HeLa cells (see Fig. 4 and Supplementary Fig. 12). Taken together, the uPEn platform demonstrates overall enhanced base-conversion efficiencies over PE3max and PE5max in multiple cell types, and presents an apparently useful tool for PE-intractable sites.

## Benchmarking the performances of uPEn against other PE platforms for insertion-, deletion- and replacement-type of edits

We further compared the efficiencies of uPEn and other major PE platforms (PE2max, PE3max, PE5maxP) in HEK293T cells for targeted

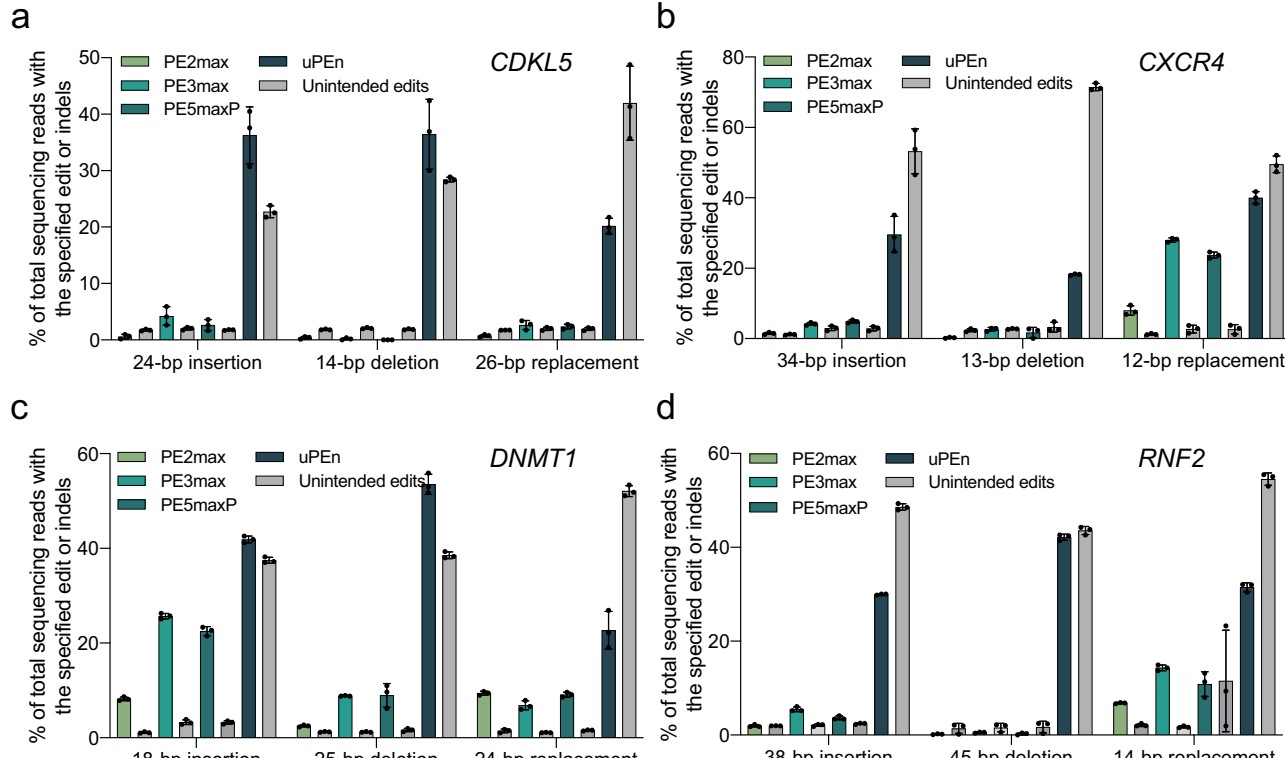

**Fig. 5 | Benchmarking the performances of uPEn for insertion, deletion and replacement. a–d** Comparison of efficiencies for insertion, deletion, replacement by PE2max, PE3max, PE5maxP and uPEn at *CDKL5* (**a**), *CXCR4* (**b**), *DNMT1* (**c**) and *RNF2* (**d**) loci in HEK293T cells. The levels of corresponding unintended edits are shown in grey bars. Values and error bars reflect the mean and s.d. of three biological replicates. Source data are provided as a Source Data file.

insertions, deletions and replacements, respectively at four endogenous sites (*CDKL5*, *CXCR4*, *DNMT1* and *RNF2*). As the intermediates for such small-block edits are more likely to evade the mismatch repair mechanisms[28], it was not surprising that here the PE5maxP groups were not associated with higher efficiencies compared to the corresponding PE3max groups. Importantly, the precise editing efficiencies by uPEn was notably higher than those by other platforms at all 4 sites (Fig. 5a–d). When editing at all sites and for all three types were summarized, uPEn empowered a median level of 34% efficiency for these small-block edits (tens of bases), while the corresponding median levels of 1.7%, 4.8% and 4.3% editing efficiencies were achieved by PE2max, PE3max and PE5maxP, respectively (Supplementary Fig. 14).

## Off-target analyses of uPEn

As uPEn adopts an active Cas9 for generation of a DSB intermediate, the potential off-target cleavage may bring some heightened safety concerns, especially when considering the low off-targeting properties of the canonical PE[4,29,30]. Therefore, we performed editing with uPEn or PEn for three different targets (i.e., sites of *HEK3*, *HEK4* and *FANCF*) in HEK293T cells, and subsequently examined their previously established off-target sites[31]. These true off-target sites (each featuring 2 to 4 nucleotide mismatches) are listed in Supplementary Fig. 15a. Following deep sequencing analyses of both on-target and off-target sites, it was found that while uPEn empowered markedly higher levels of precise editing at the target sites than PEn, the two tools induced mostly similar levels of off-targeting effects (Supplementary Fig. 15b). As the PEn/uPEn utilized in this study adopts a modified form of Cas9 (with R221K/N394K activity-improvement mutations), we further conducted whole-genome resequencing to systemically profile off-target effects by PE5max, PEn, and uPEn when the *FANCF* site was targeted for insertions. Cells transfected with EGFP-expressing plasmid were used as a control for the cells subjected to editing. The whole-genome

sequencing (WGS) was performed using genomic DNA from different groups of cells (sorted on EGFP after transfection). Sequencing was performed at depths of about 20–21× (Supplementary Fig. 16a). Variants were called based on comparisons with reference sequencing data from un-transfected cells. The results showed that cells in control, PE5max, PEn and uPEn groups featured overall similar numbers of DNA variants (Supplementary Fig. 16b). Next, potential guide RNA-dependent off-target sites with up to 5 mismatches from the target sequences were selected for further analyses. Importantly, via comparing the reads from the control and experiment groups, only one off-target edit (at a site with 2-nt mismatches, same as the corresponding "OT1" site in Supplementary Fig. 15) was identified in both the PEn and uPEn groups, but not in the PE5max group (Supplementary Fig. 16c). Therefore, the WGS data confirmed elevated levels of off-targeting by PEn or uPEn over PE5max, while suggesting no overt differences of off-targeting properties between the two PEn platforms. These results are consistent with the observations from the targeted analyses (see Supplementary Fig. 15).

Taken together, we have established an up-graded version of nuclease-based PE (uPEn) that features notably improved ratios of accurate/unintended edits over the original version of PEn. While given the relative disadvantages of imprecise on-target editing and Cas9 nuclease-associated off-target risks (compared to the canonical PE platforms), uPEn demonstrates significantly higher potencies of installing desirable edits. Our results, therefore, establish uPEn as a useful platform to complement the existing PE tools, especially when higher levels of productive edits representing a priority.

## Discussion

The majority of human pathogenic genetic variants are within the categories of nucleotide transitions or transversions, small insertions, and small deletions[32-34]. For some most severe genetic conditions, it

has been envisioned that genome editing technologies would revolutionize future therapies, eventually leading to the correction or complementation of pathogenic mutations in affected tissues/organs. Toward such an ultimate goal, the recent emergence of CRISPR-based prime editing represents a major technological breakthrough. The prime editors enable targeted, precise base changes and small insertions/deletions without the requirement of DSB[4], and therefore provide a powerful platform for future therapeutic developments and applications. Nevertheless, the efficiencies of current prime editors are generally unsatisfactory, which is likely to be associated with the unpredictable sequence of various DNA repair events associated with an initial DNA nick (by a nickase variant of Cas9) and the RT-synthesized 3'-flap[4,35]. More recently, the WT Cas9 has been adapted in the form of PE-nuclease (PEn), which appear to facilitate the installation of pegRNA-templated, RT-synthesized edits[8,10]. The convenience of not requiring another nicking sgRNA (as in the format of PE3/5) also represents an advantage for this platform. However, PEn-engaged error-prone NHEJ repair apparently caused high levels of imprecision among RT-driven edits, and induced classical indels[7,8,10,16].

Here, we aimed to develop an upgraded PEn, by seeking a 53BP1-inhibitory protein module to mitigate PEn-induced error-prone DSB repair. Such protein/PEn combination would serve as a more applicable tool, beyond the very recent proof-of-principle by the use of chemical DNA-PK/NHEJ inhibitor together with PEn[16] (also validated in Supplementary Fig. 1). We subsequently screened seven previously identified 53BP1-binding Ubv-s, using PEn-based precise insertion as a readout. Importantly, we established that a Ubv (denoted as G08) and its derivative (I44A, previously named as i53) could strongly (~2- to 5-fold, for different targets) improve the precision of installing RT-driven edits by PEn (Fig. 1b, c and Supplementary Figs. 2 and 3). Afterwards, the best-performing Ubv (i.e., i53) from the screening was then adopted for co-introduction with PEn to establish the uPEn platform. We also optimized the size of the homologous region within the RT template for effective insertions (Fig. 2b, c). To the end of insertions of tens of bps in HEK293T cells, the uPEn achieved an overall impressive efficiency of 38% (Fig. 2d). Such levels of efficiency appeared higher than those reported (ranging from 5% to 40% for 3- to 18-bp insertions) upon the application of DNA-PK inhibitor-assisted PEn[16]. Importantly, we further extended our analyses to show that uPEn also enabled small-block deletions and sequence replacements with high efficiencies [43% and 52%, respectively] (Fig. 3). Moreover, for the potentially high-demand task of precise base conversions, the uPEn markedly outperformed both PE3max and PE5max even at some largely PE-intractable sites (in HEK293T and U2OS cells, respectively), and empowered desirable base changes for overall >40% efficiencies (Fig. 4). The superior potencies by uPEn than PE3/5max for installation of small-block edits were also established (Fig. 5). Collectively, the present uPEn represent a highly efficient platform for installing a large variety of programmed, small-sized genetic modifications.

Although 53BP1 blockade (or pharmacological NHEJ inhibition) could improve conventional Cas9/template-mediated precise editing through activation of HDR[13–15,19], due to the absence of operable templates, editing by uPEn might involve a non-canonical mechanism. Our detailed analyses of allele distributions induced by PEn and uPEn, as well as those induced by applications of HR-free PEn (±NU7441), would suggest a mechanism centered on the increases of downstream DSB end resection (Supplementary Figs. 7–9). Such resection would enable correct aligning of the RT-dependent 3'-overhang to drive desirable editing. Although showing a resemblance to an end-resection-dependent, imprecise MMEJ process, the RT-based copying of HR in the present mechanism would empower the proper maintenance of genetic information for precise repair.

Such a conceived mechanism for uPEn-mediated editing is also compatible with the observation that although uPEn effectively improved the purity of the RT-driven edits in comparison to PEn, it did not mitigate the induction of classical indels (Fig. 1b). Similar results were also presented in the recent report of DNA-PK inhibitor-assisted PEn[16] (also see validations in Supplementary Fig. 1d). It may be reasoned that even with i53-stimulated resection, a naïve DSB that has not been modified a by RT-dependent upstream 3' overhang would still prone to erroneous end-joining. In this regard, the kinetics of reverse transcription of RTT following the induction of DSB might constitute the key factor that shapes the levels of direct indels by uPEn/PEn. Taken together, future works are highly warranted to comprehensively establish the DNA repair intermediates and mechanisms associated with PEn and uPEn, which shall contribute to further improving the outcomes of uPEn-enabled edits. Such explorations may also boost the outlook of adapting the uPEn platform for larger-sized edits. Moreover, although previous screening results argue against a role of 53BP1 in the regulation of PE2/3 editing[5], to further understand the molecular details contributing to the higher potencies by uPEn for installing desirable edits than canonical PE may nevertheless contribute to future optimization of the nickase-based PE platforms.

It is worth noting that our results show that diffusible, but not PEn-fusion form of i53 is evidently more effective for the enhancement of PEn (Fig. 1d, e, f). Such a condition may induce global changes in DNA damage responses, potentially causing concerns of toxicity and genetic aberrations. Nevertheless, our results showed no decreases of cell viabilities upon transient transfection of i53, either alone or together with PEn (Supplementary Fig. 6). In addition, previous studies on short-term co-administrations of i53 or other modules of 53BP1 inhibition and the CRISPR/Cas9 apparatus to the cells did not reveal increased levels of genomic damages[19], translocations events[17], or other adverse effects even in models of edited hematopoietic stem and progenitor cells monitored after in vivo reconstitution[21,22]. Although further safety evaluations are surely warranted, these lines of evidence support the feasibility of adopting i53 in transient genome editing applications. On the other hand, compared to the demonstrated high editing fidelity by canonical PE[3,4,29,30], the re-introduction of potential off-target cleavage by PEn presents some safety concerns for its application[8,10,16]. It is interesting to note that the uPEn exhibited mostly similar levels of off-target editing as the PEn (Supplementary Figs. 15, 16), consistent with a notion that 53BP1 inhibition does not affect Cas9-mediated DNA cleavage. Future adoption of high-fidelity Cas9s or optimization of guide RNA structures[3] may present viable strategies to improve uPEn's safety profiles.

In spite of certain considerations for future improvements, the current version of uPEn has presented a broadly efficient and versatile prime editing platform. It would be readily suitable for editing cells or animal models where high-potency prime editing is needed and the precisely modified population may be selected. It also represents a useful tool that may potentially be applied for the correction of pathogenic mutations where the target alleles cause loss-of-functions. Indeed, the possibility of genetically correcting a percentage of cells is relevant to the treatment of conditions such as certain severe blood disorders or muscular dystrophy[1]. The uPEn platform holds strong potential for future developments and applications.

## Methods

### Cell culture, transfection, and harvest

HEK293T (ATCC CRL-3216), U2OS (ATCC HTB-96), and HeLa (ATCC CCL-2) cell lines were cultured in Dulbecco's Modified Eagle's Medium supplemented with 10% (v/v) fetal bovine serum (FBS) at 37 °C with 5% CO2. For plasmid transfection, cells were seeded in 24-well a day before and transfected at about 70% density using lipofectamine 2000 (Thermo Fisher Scientific) twice [2 μl/μg] the amount of plasmid DNA as per the manufacturer's instructions. For PEn experiments, cells were transfected with 900 ng of PEn plasmids and 300 ng of pegRNA plasmids. For uPEn1 experiments, cells were transfected with 900 ng of PEn plasmids, 300 ng of pegRNA plasmids and 300 ng of ubiquitin

variants plasmids. For uPEn2 and uPEn3 experiments, cells were transfected with 900 ng of uPEn2 or uPEn3 plasmids and 300 ng of pegRNA plasmids. For PE3max and PE5max experiments, cells were transfected with 900 ng of PE2max plasmids, 300 ng of pegRNA plasmids and 100 ng nick-sgRNA plasmids (PE5max with extra 450 ng hMLH1dn plasmids). In Supplementary Figs. 13 and 14, we used PEmax-P2A-hMLH1dn plasmid for the PE5maxP groups. Cells were transfected with 900 ng of PEmax-P2A-hMLH1dn plasmids and 300 ng 2U6 plasmids (PE5maxP-2U6), or with 900 ng of PEmax-P2A-hMLH1dn plasmids, 300 ng of pegRNA plasmids and 100 ng nick-sgRNA plasmids (PE5maxP). Three days after transfection, cells were harvested from Fluorescence Activating Cell Sorter (FACS) with EGFP$^+$ (or EGFP$^+$/mCherry$^+$) selection.

## Flow cytometry

Three days after transfection, cells were harvested for flow cytometry sorting (BD Aria III, programmed via the FACSDiva (8.0.1) software). Cell sorting is based on the EGFP marker on the pegRNA plasmids, and sometimes also on the mCherry marker on the free Ubv plasmids or the nick-sgRNA plasmids. When comparing groups indicated by either a single EGFP marker, or by both EGFP and mCherry markers, an EGFP$^+$ gate or an EGFP$^+$mCherry$^+$ gate (both gates with the same range of FL1-fluorescence) was respectively used for sorting. The gating strategy is presented using representative dot plots (Supplementary Fig. 17a, b). A total of 10,000 positive cells were collected by FACS for subsequent genomic DNA preparation.

## Plasmid construction

For the sgRNA construction, pegRNA plasmid was constructed according to the methods described in our previous study[36]. The detailed methods are as specified next. To construct pegRNA plasmids, the plasmid backbone was amplified from pGL3-U6-sgRNA-EGFP (Addgene, #107721) using Phanta® Max Super Fidelity DNA Polymerase (Vazyme). The backbone amplicon was then cut by BsaI-HFv2 (NEB) for overhangs. Spacer oligos of pegRNAs (the top strand oligo includes 5′ ACCG and 3′ GTTTT overhangs, while the bottom strand oligo comprises a 5′ CTCTAAAAC overhang), pegRNA 3′ extension (the top strand oligo included 5′ GTGC overhang while the bottom strand oligo included 5′ AAAA overhang), and sgRNA scaffold oligos (featuring compatible overhangs) were synthesized. Next, the four fragments are assembled with T4 DNA Ligase (NEB). General primers used for constructions are listed in Supplementary Table 1. Sequences for pegRNAs used to program fragmental modifications (HR-pegRNAs) are listed in Supplementary Data 1. Sequences for pegRNAs for comparisons with PE2max, PE3max or PE5max are listed also in Supplementary Data 1. For conventional Cas9/template-based editing, the ssDNA templates were synthesized. Their sequences are listed in Supplementary Table 2.

For PEn construction, PEmax prime editor plasmid was purchased from Addgene (Addgene, #174820), oligos were synthesized and the H840A Cas9 mutation in the PEmax construct was reverted to the original histidine. For ubiquitin variants expression plasmids, sequences encoding seven ubiquitin variants were synthesized by GeneScript, after which they were amplified by PCR and cloned into the pEF1a vector to generate the ubiquitin variants-encoding plasmids. For uPEn2 and uPEn3 construction, sequences encoding 34aa linker and P2A were synthesized by GeneScript and UbvG08(I44A) were amplified by PCR, they were cloned into PEn plasmid to generate uPEn2 and uPEn3 plasmids, respectively. Complete sequences for some key constructs are listed in the section of Supplementary Notes.

For Flag-i53 plasmids construction, paired primers corresponding to the 3xFlag sequence were synthesized by GeneScript. The 3xFlag sequence was then cloned into the C-terminus of UbvG08 (I44A) by PCR to generate the Flag-i53 plasmids. For the construction of plasmids containing two U6-guide RNA cassettes, the sequenced, including the U6 promoter, the spacer region, and the scaffold of nick-sgRNA were amplified by PCR, and subsequently cloned downstream of the U6-pegRNA cassette in the pegRNA plasmids.

## Cell viability assay

Cell viability assay was performed according to the manufacturer's instructions (CellTiter-Lumi™ Plus Luminescent Cell Viability Assay Kit, Beyotime, Shanghai, China). 72 h after transfection, an equal volume of CellTiter-Lumi™ Plus Reagent was added to the wells. The samples were mixed for 2 min at room temperature on an orbital shaker to induce cell lysis. After an additional 10 min incubation at room temperature to stabilize the luminescent signal, the samples were subjected to measurements by a luminometer.

## Western blotting

For Western blotting, after plasmid transfection, HEK293T cells were harvested and lysed in RIPA buffer. The blots were subjected to incubations with anti-Cas9 (Abcam ab204448, 1:2000), anti-β-actin (Absin abs132001, 1:10000), or anti-Flag (Abcam ab205606, EPR20018-251, 1:5000) antibodies in TBST with 5% skim milk. Images were captured with Amersham Imager 600. The original scanned blots for the presented Western results are included in the Source Data file.

## RT−qPCR of pegRNAs

Transfection of HEK293T with PE2max, PEn and uPEn plasmids (with pegRNA) was performed as described above. Total RNA from transfected cells was isolated using the RNA isolater Total RNA Extraction Reagent (Vazyme). The HiScript Q RT SuperMix for qPCR [+gDNA wiper] (Vazyme) was used to generate cDNA using random hexamers. The qPCR was carried out with primers corresponding to the spacer and the 3′ extension sequences of the pegRNA, using a commercial reaction mix from Vazyme (AceQ qPCR SYBR Green Master Mix [Low ROX Premixed]). The pegRNA signals was normalized to those of the transcripts corresponding to Cas9 sequence. Fold changes in mRNA abundances were calculated using the 2-ΔΔCt method. Primer sequences are available in Supplementary Table 3.

## Genomic DNA extraction and genotyping

The genomic DNA was extracted using QuickExtract™ DNA Extraction Solution (Lucigen) according to manufacturer's protocols. The isolated DNA was PCR-amplified with Phanta® Max Super-Fidelity DNA Polymerase (Vazyme). Primers used are listed in Supplementary Data 1.

## Targeted deep-sequencing

The harvested genomic DNA samples were subjected to PCR amplification and targeted deep-sequencing[37] for the on-target and off-target genomic sites. For analyses of the same sites upon different experimental conditions, the 5′-primers with distinct barcodes were used for preparation of the amplicons. The primers used for the targeted amplicons are listed in Supplementary Data 1. The primers were designed so that the nCas9 cleavage sites would be located close to the center of the amplicons. The target sites were amplified with Phanta® Max Super-Fidelity DNA Polymerase (Vazyme). Touchdown PCR reactions proceeded for 35 cycles. PCR products with different barcodes were pooled together. The biological replicates were respectively pooled into different sequencing samples. The gel-purified PCR products were subjected to end repair and adaptor ligation according to manufacturers' instructions (Illumina). The deep sequencing experiments were carried out using the Illumina NovaSeq platform (PE150 and PE250) at Genewiz. Sequencing reads were demultiplexed using AdapterRemoval (v.2.2.2), and the pair-end reads with 11 bp or more alignments were combined into a single consensus read. All processed reads were then mapped to the target sequences using the BWA-MEM algorithm (BWA v.0.7.17). Levels of accurate edits were calculated as the percentages of reads with the desired edits (no extra indels) in total

mapped reads. Levels of "All RT-driven edits" were calculated as percentages of reads with any RT-dependent edits in total mapped reads. Levels of direct indels were calculated as the percentages of reads containing classical NHEJ indels in total mapped reads. Sometimes, levels of imprecisely repaired edits (all RT-driven edits−accurate edits) were also determined for presentation purposes. To present the levels of all "Unintended edits", the percentages of direct indels + imprecisely repaired edits were determined. For some experiments, the demultiplexed pair-ended reads were subjected to analyses by CRISPResso2 under the HDR mode to generate explicit demonstrations of allele distributions. The indel quantification windows were set to extend 30-bp beyond the positions of the cleavage site and the end position of the template-dependent RT. For sequence alignment, the plot_window_size was adjusted according to the combined sizes of the insertion and the HR.

### Off-target analysis

We performed off-target analyses for three previously profiled spacer sequences[31]. The region around off-target sites were amplified with Phanta® Max Super-Fidelity DNA Polymerase (Vazyme), and subjected to high-throughput sequencing with using Illumina NovaSeq (PE150). Sequencing reads were demultiplexed using AdapterRemoval (v.2.2.2), and the pair-end reads with 11 bp or more alignments were combined into a single consensus read. All processed reads were then mapped to the target sequences using the BWA-MEM algorithm (BWA v.0.7.17). The off-target sites are listed in Supplementary Fig. 15. Primers used are listed in Supplementary Data 1.

### Whole-genome sequencing

DNA extracted from harvested cells was sequenced using Illumina NovaSeq (PE150) at the Annoroad Gene Technology, Beijing, China. All cleaned reads were mapped to the human reference genome (GRCh38/hg38) using BWA v.0.7.17 with default parameters. Sequence reads were removed for duplicates using Sambamba v.0.6.7. Variants were identified by GATK (v.4.1.8.1) HaplotypeCaller and filtered with the following criteria: (1) sequencing depth (for each individual) >1/3× and <3×; (2) variant confidence/quality by depth >2; (3) RMS mapping quality (MQ) > 40.0; (4) Phred-scaled $P$ value using Fisher's exact test to detect strand bias <60; (5) Z-score from the Wilcoxon rank sum test of Alt vs. Ref read MQs (MQRankSum) >−12.5; and (6) Z-score from the Wilcoxon rank sum test of Alt vs. Ref read position bias (ReadPosRankSum) >−8. Reference sequencing data from un-transfected cells were used to filter pre-existing mutations. Potential off-target sites were predicted by Cas-OFFinder[38].

### Statistics and reproducibility

No statistical method was used to predetermine the sample size. No data were excluded from the analyses. The study only involved the use of established mammalian cell lines, with each cell line cultured and handled under an identical condition. The experiments were conducted in a per cell-line basis. Therefore, the experiments were not randomized. As the measurements in the study were objective, the Investigators were not blinded to allocation during experiments and outcome assessment. All quantitative sample-measurements were conducted with three biological replicates. Sometimes, results from the use of a given editing platform on multiple sites were summarized together to provide a generalized view of the findings. Graphpad prism v.9 was used to analyze the data (v.9.1.1). Mean values and standard deviations are displayed. Students' $t$-test (two-tailed) were used to determine the statistical significance of differences in levels among experimental groups.

### Reporting summary

Further information on research design is available in the Nature Portfolio Reporting Summary linked to this article.

## Data availability

Targeted amplicon sequencing and WGS data have been deposited to the NCBI-SRA repository under BioProject number: PRJNA847383. Descriptions of the treatments and samples included in the dataset are provided in Supplementary Data 1 (in a sheet named "SRA"). The reference human genome assembly GRCh38/hg38 used for reads mapping is an openly accessible resource (https://www.ncbi.nlm.nih.gov/assembly/GCF_000001405.40). Source data are provided with this paper.

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

## Acknowledgements

We thank members of Huang lab for helpful discussions and thank the Molecular and Cell Biology Core Facility (MCBCF) at the School of Life Science and Technology, ShanghaiTech University, for providing technical support. Data analysis is supported by the HPC Platform of ShanghaiTech University. This work is supported by the National Key R & D Program of China (2021YFF1000704 to J.L., 2021YFA0804702 to X.H.), the Key Research Project of Zhejiang Laboratory (2021PE0AC06 to X.H. and S.Z.), and the Leading Talents of Guangdong Province Program (2016LJ06S386 to X.H.).

## Author contributions

X.H., S.Z., J.L., X.W., Q.J., and Y.Y. conceived, designed, and supervised the project. X.L., G.Z., and S.H. performed most experiments with the help of Y.L., W.S., and X.W. M.Z. provided expert technical assistance. X.L. and G.Z. wrote the paper with inputs from all authors. X.H., S.Z., J.L., X.W., Q.J., and Y.Y. revised the manuscript and managed the project.

## Competing interests

The authors declare no competing interests.

## Additional information

**Peer review information** : *Nature Communications* thanks the anonymous reviewer(s) for their contribution to the peer review of this work. Peer reviewer reports are available.

