## [Peer Review File · Nature Communications]

Reviewers' Comments:

Reviewer #1:

Remarks to the Author:

Verdict: Reject

In this manuscript, Li and colleagues iterate on nuclease/prime editing technology. The authors claim:

1. PE nucleases are improved by the addition of a 53BP1 inhibitor.
2. This improves editing precision over the original PE nuclease.
3. pegRNA parameters can be optimized to improve editing outcomes.
4. PE nuclease systems can edit loci that are refractory to other PE approaches.

Overall, the manuscript is well written, and data/figures are clear. The manuscript shows that a combinatorial approach to gene editing can boost editing relative to another "prime-editor". The authors are resourceful in utilizing new advances from literature, noticing that a small molecule DNA-PK (AZD7648) can boost HDR rates in Cas9 edited cells by inhibiting NHEJ. They utilize another approach in the manuscript (i53) that involves a mutated ubiquitin that binds 53BP1 preventing participation in NHEJ at editing sites. This addition increases editing relative to PEn by itself. They then modify the reverse transcribed template cDNA by adding a downstream homology region pegRNA, which also improves editing. I felt that this work optimizing pegRNA design was a high point of the manuscript.

However, the overall impact of this manuscript was difficult to define. The authors have taken two separate types of gene editing and merged them. There is little discussion about the relative tradeoffs of each approach (cutting versus prime editing) or mechanistic insights into why techniques that improve cutting-based editing works also boosts editing with prime editors (or prime editing nucleases). I spent my time reading this manuscript wondering three things: 1) under what circumstances would I want to use a prime editing nuclease in my own work, 2) why does this approach work mechanistically, and 3) what logic did the authors follow to develop this approach? This latter question is a big one: the whole point of prime editing is to avoid DSBs. The authors reintroduce the drawbacks of cutting editing, such as increased indel frequency, for modest increases in DNA repair. They do not make a sufficient case that the benefits outweigh the drawbacks.

Major concerns:

1. Comparisons made within figures are incomplete. This is a hybrid gene editing approach. The authors should compare their approaches both to Cas9 cutting and to current prime editing reagents. This is done incompletely in the current form of the manuscript. For example, in figure 1, PEn/uPEn are arrayed and compared at various sites. Adding cutting Cas9/template and one of the original PE variants 2/3/5 would greatly clarify relative performance. As it is, it looks like PE nucleases perform slightly better than the PE reagents in the Anzalone et al manuscript but with greatly increased indel frequencies. It's common for editing reagents to work less efficiently than advertised, so I would not ask the authors to repeat efficiencies from another manuscript, but they need to show me that PEns work better than PE reagents in their hands.

2. There is little mechanistic insight. Why do PE nucleases work? Why does i53 addition work? The authors have combined a few gene editing approaches, reported an increase in efficiency, and moved on. The authors should make some effort to define why their interventions work. Maybe all this works through 53BP1, but I can think of trivial options not discussed by the authors: 1) maybe the fusion of the RT to Cas9 stabilizes the nuclease or increases its production, or 2) maybe pegRNA templates are protected or imported more effectively because they are packaged in the nuclease. It's essential for progress in the field that these editing interventions have some mechanism behind them. The authors need to develop hypotheses about how PEn/uPEn systems

work and test them.

3. Benefits of PE nucleases are unclear. Compared to the original Prime editing paper by Anzalone et al. from David Liu's lab, uPEn only has marginally improved editing rates at the cost of significantly increased indels due to DSB activity instead of nickase. This appears to be a major drawback. Off-targets were a major concern original of Cas9 DSB induced editing. This was reduced by using a nickase variant for PE, but PEn reintroduces this. Does uPEn have reduced OT activity vs. PE or PEn? Bioinformatically determined OT sites are often insufficient, do the authors have work showing that biologically determined OT sites are less so affected?

4. Side effects of i53 are not properly monitored. The original suggestion that NHEJ inhibition by small molecule DNA-PK (AZD7648) has non-specific effects and may present toxicity effects is a valid concern. However, the authors do not show whether their i53 ubiquitin variant is toxic. Furthermore, the results suggest that this ubiquitin variant works best with cell-wide administration (i.e. uPEn variant 1 and 3). Is overexpression of this ubiquitin variant in the nucleus inhibiting NHEJ or sequestering 53BP1 in a non-reversible manner? Is an increase in general DNA damage/toxicity in treated cells? Is there an increased probability of neoplastic events?

Reviewer #2:

Remarks to the Author:

The ability of CRISPR prime editing to efficiently introduce precise editing including small indels and substitutions has made this technology to become the next generation gene editing technique. Prime editing is originally derived from the fusion of Cas9-nickase and reverse-transcriptase. As of now, several strategies and novel improvements to use this technology have been developed. PE2 strategy relies on single gRNA and single-strand break therefore avoiding DNA double strand break. However, PE2 efficiency is low. Additional gRNA targeting the opposite strand (second nick) enhances the prime editing efficiency, a technique dubbed as PE3 which basically creates a staggered-DSB similar to a dual-nickase approach. Optimization of the codon usage and NLS has created PEmax. PE3 with this optimization is called PE3-Max and addition of MMR inhibitor which is called PE5-Max was claimed to improve prime editing.

To bypass the need of the second-nick gRNA for enhanced prime editing, Adikusuma et al generated prime editing nuclease. Although prime editing initiation was robust, it suffered from unwanted prime editing outcomes whereas the transcribed homology undergoes non-homologous end joining (NHEJ) instead of strand annealing.

In this study Li et al cleverly used NHEJ inhibitor called i53 discovered by Canny et al that can inhibit the key player of NHEJ, the 53BP1. They found that this strategy could improve accurate prime editing as opposed to unwanted template insertions as mostly seen in the standard nuclease prime editor.

This study is interesting and the finding is important in the field of CRISPR gene editing. This finding will contribute significantly to improve and simplify prime editing technique.

However, there are some concerns that I would like the authors to address:

The NHEJ inhibitor factors used in this study was based on Canny et al (Nat Biotech 2019). Canny et al tested several candidates and found UbvG08(I44A) as the best 53BP1 inhibitor and dubbed this factor as "i53". However authors did not mention i53 at all.

Line 77: Authors stated the uncertainty of specificity/toxicity of small molecule inhibitors. However, authors could not provide evidence the harmless of overexpression of the UbvG08 factors.

Fig 1E: Although I understand this graph, other readers might get confused with the 'indel' bars.

Introduction: It is better if the authors remind the readers regarding PE2, PE3 and PE5 systems as

well as reminding that the PE nuclease uses only a single spacer instead of dual-guides as in PE3 or PE5.

Line 120: Adikusuma et al (NAR 2021) observed high prime editing event but mostly resulted in unwanted template insertions (from PEn). It is unclear in this study whether improved accurate prime editing efficiency is due to the reduction of unwanted templated insertions that are converted to correct prime editing. If it is the case, authors need to make a bold statement and make it clear in the figure (Fig 1) by including the data showing the frequency of the unwanted template insertions.

Line 144 and Figure 1E: It is important to include the frequency of unwanted template insertions to emphasize that increased PE was due to the conversion of unwanted template insertions to correct PE. This data is crucial and I expect this important observation is also seen across multiple cell lines.

Line 130: given no experimental comparison was performed, I suggest moving this statement to Discussion.

All data (including supplementary) should show individual values by using scattered plot bar.

Figure 1B is a representative of 1 of 3 sites performed by authors. It would be better to see all 3 sites in one graph. To save some space I would suggest the graph to contain only control, Ub (WT), G08 and G08(I44A) and move the remaining to the supplementary data.

Line 157: It is too immature to draw strong conclusion of the optimal length of the homology arms based on this experiment given that only insertions of >24 bp were performed. Therefore I would suggest toning down the sub-heading e.g. "optimizing the size of....." to indicate that this optimal size might be relevant only for this study rather than for general use.

Figure 2D: The low efficiency in PAH and FANCF could be caused by longer inserts? Can be mentioned in the text.

Line 172: chose instead of choose.

Line 181 & 211: Worth mentioning that uPEn-3 system was used.

Line 184: Six genomic loci however there are only 3 apparently.

The uPEn system was not tested in hard-to-prime-edit cells such as HeLa.

Line 203-238 and Material & Method: In this study, authors generated the PEn and uPEn using the backbone of PE-Max which uses optimized codon and has extra NLS. However, when comparing the uPEn with PE3 and PE5, authors used PE2 construct for the PE3 and PE5 systems that doesn't have optimized codon and extra NLS. To me this is the major flaw of this study. To enable fair comparison, authors should use PE2-Max construct for the comparison.

Methods: the amount of lipofectamine was not mentioned.

Methods: uPEn transfection ◊ 900 ng uPEn plasmid + 300 ng pegRNA plasmid
PE3 transfection ◊ 900 ng PE2 + 300 ng pegRNA + 100 ng nick gRNA plasmid
PE5 transfection ◊ 900 ng PE2 + 300 ng pegRNA + 100 ng nick gRNA + 450 ng hMLH1dn plasmid
In addition to using PE2-Max construct, I would like to point out the possibility of inefficient transfection of the extra plasmids (nick gRNA + hMLH1dn plasmids). It would be a more fair comparison if the nick gRNA is expressed in the same plasmid as the pegRNA.
For comparison with PE5, plasmid PE-Max-P2A-hMLH1dn <https://www.addgene.org/174828/> should be used. This will rule out the possibility that inefficiency of PE3/PE5 is caused by inefficient transfection due to the presence of extra plasmids.

Comparison with PE3max and PE5max should also perform editing the same as previously targeted

by David Liu.

Generated plasmids are expected to be available to the community through Addgene.

In summary, this manuscript requires major reconstruction and considerably more experimental data to confirm the findings and draw reliable conclusions.

MS title: Development of a versatile nuclease prime editor with upgraded precision, uPEn

Author: Li, et al.,

MS number: NCOMMS-22-23029

We are very grateful to the reviewers for their insightful comments and suggestions. We thank the reviewers for having pointed out the positives, as well as certain insufficiencies in our work. Some of the concerns may have been ascribed to our inadequate elaboration of the study rationale, for which we sincerely apologize.

We have since performed additional experiments/analyses and duly revised our manuscript to address all the points raised by the reviewers. Please also see separately attached files for a new version of manuscript (major revisions highlighted). Our point-by-point responses are detailed below.

Responses to the reviewer's comments:

Reviewer #1

In this manuscript, Li and colleagues iterate on nuclease/prime editing technology.

The authors claim:

1. PE nucleases are improved by the addition of a 53BP1 inhibitor.
2. This improves editing precision over the original PE nuclease.
3. pegRNA parameters can be optimized to improve editing outcomes.
4. PE nuclease systems can edit loci that are refractory to other PE approaches.

Overall, the manuscript is well written, and data/figures are clear. The manuscript shows that a combinatorial approach to gene editing can boost editing relative to another "prime-editor". The authors are resourceful in utilizing new advances from literature, noticing that a small molecule DNA-PK (AZD7648) can boost HDR rates in Cas9 edited cells by inhibiting NHEJ. They utilize another approach in the manuscript (i53) that involves a mutated ubiquitin that binds 53BP1 preventing participation in NHEJ at editing sites. This addition increases editing relative to PEn by itself. They then modify the reverse transcribed template cDNA by adding a downstream homology region pegRNA, which also improves editing. I felt that this work optimizing pegRNA design was a high point of the manuscript.

However, the overall impact of this manuscript was difficult to define. The authors have taken two separate types of gene editing and merged them. There is little discussion about the relative tradeoffs of each approach (cutting versus prime editing) or mechanistic insights into why techniques that improve cutting-based editing works also boosts editing with prime editors (or prime editing nucleases). I spent my time reading this manuscript wondering three things: 1) under what circumstances would I want to use a prime editing nuclease in my own work, 2) why does this approach work mechanistically, and 3) what logic did the authors follow to develop this approach? This latter question is a big one: the whole point of prime editing is to avoid DSBs. The authors reintroduce the drawbacks of cutting editing, such as

increased indel frequency, for modest increases in DNA repair. They do not make a sufficient case that the benefits outweigh the drawbacks.

[Authors' responses]

We thank the reviewer very much for these comments on the general positives and insufficiencies of our manuscript. Regarding the reviewer's uncertainties on the overall logic of this investigation, we sincerely apologize for our inadequate elaborations in the initial manuscript. Some further clarifications are provided below.

The reason that PEn represents an interesting platform for further explorations:

We concur with the reviewer that one of the important features of the canonical PE systems is their activities to mediate precise genome editing independent of generation of DSBs. Nevertheless, the existing PEs show suboptimal and inconsistent activities, despite significant optimization efforts by different groups. It is widely recognized that various DNA repair pathways play intricate roles in shaping the outcomes of PE¹. Therefore, efforts to rationally manipulate the editing intermediates and/or repair pathways represent a major direction for development of improved versions of PEs, or those with specialized activities. We revised the Introduction to further elaborate such points (line 70-74).

The very recent development of PE-nuclease (PEn) platforms, via the use of WT Cas9-RT fusion protein and a pegRNA², provided some important insights. Although this tool is expectedly associated with error-prone editing, a fraction of the products features precise edits (at levels sometimes higher than the canonical PE). Interestingly, besides the Cas9-dependent direct indels, the majority of unwanted edits showed imprecise/partial duplication of the RT template sequence at the target site². Therefore, in spite of mainly engaging an error-prone pathway, PEn leads to preferred acquisition of intermediates containing reverse-transcribed DNA for subsequent DSB resolution. Such a notable ability by PEn (more potently than the canonical PE) to drive RT-dependent edits provides the basis for its potential improvements toward a high-activity genome editing tool. Related texts involving the reported developments of PEn have been added to the Introduction and to the first paragraph of the Results (line 75-86; line 115-122).

The rationale for uPEn development and the advantages of this platform:

Targeting the error-prone NHEJ pathway in favor of the homology-dependent repair (HDR) has previously been used for improving traditional Cas9/donor template-dependent editing, as the reviewer has also kindly noted. Herein, given the prevalence of RT-dependent, end-joining type imprecise edits by PEn, we hypothesized that similar strategies of manipulating the DSB repair could be adopted to drive precise edits at the expense of unintended edits. The reasoning and background for targeting NHEJ in PEn are revised for a better logic flow in the Introduction (line 81-92).

Based on its role in promoting NHEJ repair of DSBs, 53BP1 protein represents a potential target to promote PEn-dependent precise editing outcomes (mentioned in the original manuscript). Therefore, the present work revolved around combining PEn with

a 53BP1-inhibitory, engineered protein. Further optimization of the system led to the development of a uPE system that enabled efficient installation of the desirable RT-dependent edits with high purity (marked improvements over PE). Furthermore, when the absolute levels of precise editing were considered, this upgraded platform of PE also substantially out-performed the canonical PE platforms (with Cas9 nickase). Therefore, despite the caveats of inducing certain levels of collateral Cas9-dependent indels and other potential complications, uPE's strong activity to install precise RT-dependent edits may possibly outweigh its downsides. We believe that uPE would be especially suited for applications that may tolerate certain levels of imprecise editing, or those that include selection steps (mentioned originally in the last paragraph). Texts commenting on the technical advantages of the current uPE platform are added to the Results (line 250-257; 493-500).

Major concerns:

1. Comparisons made within figures are incomplete. This is a hybrid gene editing approach. The authors should compare their approaches both to Cas9 cutting and to current prime editing reagents. This is done incompletely in the current form of the manuscript. For example, in figure 1, PE/uPE are arrayed and compared at various sites. Adding cutting Cas9/template and one of the original PE variants 2/3/5 would greatly clarify relative performance. As it is, it looks like PE nucleases perform slightly better than the PE reagents in the Anzalone et al manuscript but with greatly increased indel frequencies. It's common for editing reagents to work less efficiently than advertised, so I would not ask the authors to repeat efficiencies from another manuscript, but they need to show me that PEs work better than PE reagents in their hands.

[Authors' Responses]

We thank the reviewer for this important suggestion. Such comparisons would underscore the potential advantages for the nuclease-based PE platform compared to the canonical PE. We have carried out a number of additional experiments in this regard. Some related results shown in the original manuscript are also described below to support the overall conclusion.

Firstly, we compared the editing efficiencies of PE/uPE with those of the conventional Cas9/ssDNA donor strategy and of PEmax (PE2 format, "PE2max") for installing 18-bp insertion at three endogenous sites in 293T cells. For the conventional editing strategy, we designed two ssDNA templates. One ssDNA template (ssDNA1) adopted a similar structure as the 3' extension of pegRNA, which was composed of the sequences corresponding to PBS and RT template (3' to 5'). In contrast, the other ssDNA template (ssDNA2) was designed as a standard donor ssDNA with respective 35-bp homologous arms flanking the sequence of insertion. As expected from the suboptimal size of the homologous arms in ssDNA1 (20- and 13-bp, respectively) for engagement of HDR, the Cas9/ssDNA1 editing format resulted in the lowest efficiency of precise editing among all groups (see **Rebuttal Figure 1**). The unmodified PE induced equivalent levels of precise edits as the standard Cas9/ssDNA2 editing format.

On the other hand, the use of PE2max led to overall higher levels of precise editing than PEn. Notably, uPEn not only substantially favored accurate edits over unintended edits in comparison to the PEn and Cas9/ssDNA groups, but also drove the highest levels of precise edits among all groups (see **Rebuttal Figure 1**). These results underscored the advantages by uPEn in its combined capabilities of engaging pegRNA-dependent reverse transcription and of potentiating an error-free DSB repair pathway(s), therefore empowering efficient precise editing. Although the use of uPEn was still associated with higher levels of unintended edits as compared to PE2max, it would be reasonable to perceive that its markedly enhanced efficiencies for installing accurate edits might outweigh such a setback, at least under certain application contexts with the editing efficiency as priorities. These results have been added to the revised manuscript (Supplementary Fig. 5, line 222-257).

In the original version of the manuscript, following the extensive comparisons between uPEn and PEn, we also carried out editing experiments for base conversions using uPEn in parallel to PE3max [HEK293T] or PE5max [U2OS] (current Fig. 4 and Supplementary Fig. 11). The results showed that while these examined sites were largely refractory to PE3/5max, their targeting by uPEn led to notable levels of desirable edits.

Furthermore, for targeted insertions, deletions and replacements, we benchmarked the performances by uPEn against other major PE platforms (PE2max, PE3max, PE5max) in HEK293T cells (see **Rebuttal Figure 2**). As the intermediates for such block-type edits are more likely to evade the mismatch repair mechanisms, it was not surprising that here the PE5max groups were not associated with higher efficiencies compared to the corresponding PE3max groups. Importantly, the precise editing efficiencies by uPEn was notably higher than those by other platforms at all 4 sites. These results have been added to the revised manuscript (Supplementary Fig. 14 and line 449-460).

Taken together, our results demonstrate the generally higher potencies by uPEn than the canonical PE platforms to install a variety of prime edits, despite the relative disadvantages of imprecise editing. We therefore establish uPEn as a useful platform to complement the existing PE tools, especially when higher levels of productive edits representing a priority (mentioned in Result and Discussions, line 250-257; 493-500; 604-606).

Rebuttal Figure 1. The editing efficiencies of PEn/uPEn compared to Cas9/template and PE2max system.

a. Comparison of 18-bp insertion efficiencies induced by Cas9/ssDNA, PE2max, PEn and uPEn at the *SEC61B* site in HEK293T cells. Note that ssDNA1 was a pegRNA 3' extension-like sequence, including PBS sequence, RT template sequence; ssDNA2 was a standard donor ssDNA with 35-bp homologous regions flanking the designed insertional sequence. Values and error bars reflect the means and standard deviation (s.d.) of three biological replicates.

b. Comparison of 18-bp insertion efficiencies induced by Cas9/ssDNA, PE2max, PEn and uPEn at the *RUNX1* site in HEK293T cells. Values and error bars reflect the means and standard deviation (s.d.) of three biological replicates.

c. Comparison of 18-bp insertion efficiencies induced by Cas9/ssDNA, PE2max, PEn and uPEn at the *LSP1* site in HEK293T cells. Values and error bars reflect the means and standard deviation (s.d.) of three biological replicates.

Rebuttal Figure 2. Benchmarking the performances of uPEn for insertion, deletion and replacement.

- a. Comparison of efficiencies for insertion, deletion, replacement and indel rates by PE2max, PE3max, PE5max and uPEn at *CDLK5* loci in HEK293T cells. Values and error bars reflect the mean and s.d. of three biological replicates.
- b. Comparison of efficiencies for insertion, deletion, replacement and indel rates by PE2max, PE3max, PE5max and uPEn at *CXCR4* loci in HEK293T cells. Values and error bars reflect the mean and s.d. of three biological replicates.
- c. Comparison of efficiencies for insertion, deletion, replacement and indel rates by PE2max, PE3max, PE5max and uPEn at *DNMT1* loci in HEK293T cells. Values and error bars reflect the mean and s.d. of three biological replicates.
- d. Comparison of efficiencies for insertion, deletion, replacement and indel rates by PE2max, PE3max, PE5max and uPEn at *RNF2* loci in HEK293T cells. Values and error bars reflect the mean and s.d. of three biological replicates.
- e. Statistical analyses on the efficiencies of targeted insertion, deletion and replacement by PE2max, PE3max, PE5max and uPEn system in HEK293T cells. Boxes represent 25th–75th percentile (line at the median). $n = 12$ for each group. P values were calculated by a two-tailed Student's t-test.

2. There is little mechanistic insight. Why do PE nucleases work? Why does i53 addition work? The authors have combined a few gene editing approaches, reported an increase in efficiency, and moved on. The authors should make some effort to define why their interventions work. Maybe all this works through 53BP1, but I can think of trivial options not discussed by the authors: 1) maybe the fusion of the RT to Cas9 stabilizes the nuclease or increases its production, or 2) maybe pegRNA templates are protected or imported more effectively because they are packaged in the nuclease. It's essential for progress in the field that these editing interventions have some mechanism behind them. The authors need to develop hypotheses about how PEn/uPEn systems work and test them.

[Authors' Responses]

We thank the reviewer for asking about the mechanisms underlying the improved potencies for installation of desired edits by uPEn, and about the differences among various editing approaches. We apologize again for having not provided the rationale of pursuing the PEn platform, which may have left the reviewer uncertain about the editing features by PEn.

(I) Differences in editing mechanisms between PE and PEn:

The mechanisms for PE have been studied for more details (descriptions included in the original version). Nevertheless, the existing PEs show suboptimal and inconsistent activities, despite significant optimization efforts by different groups. Efforts to manipulate the editing intermediates and/or repair pathways represent a

direction for development of improved versions of PEs, or those with specialized activities.

The very recent development of PEn platforms provided some important insights. Although this tool is expectedly associated with error-prone editing, a fraction of the products features precise edits (at levels sometimes higher than the canonical PE). However, the initial works on PEn did not reveal the mechanisms for the fraction of desirable edits². Interestingly, besides the Cas9-dependent direct indels, the majority of unwanted edits showed imprecise/partial duplication of the RT template sequence at the target site, characteristic of NHEJ. Therefore, in spite of mainly engaging an error-prone pathway, PEn leads to potent acquisition of intermediates containing reverse-transcribed DNA for subsequent DSB resolution. Therefore, inhibition of NHEJ had been conceived for improvement of PEn, which formed the basis for the work by Peterka et al.³ on adopting a DNA-PK inhibitor. Related texts are now provided in the revised manuscript (line 73-92, 115-122).

We also performed the PEn and DNA-PK inhibitor-assisted PEn experiments targeting a series of sites. However, as such analyses have been reported by Peterka et al. very recently, we had showed this part only in the cover letter to Editor upon the first submission (see **Rebuttal Figure 3**). Now for clarity purposes, we have added these results to the manuscript. Our experiments focused on another set of target sites and on using a different DNA-PK inhibitor. We found that although application of PEn generally resulted in moderate levels of accurate edits, the large majority of the unintended edits indeed featured partial duplication of RT template sequences (**Rebuttal Figure 3A**). The WT Cas9-dependent straight indels were also apparently induced by PEn. Such patterns were consistently observed across all sites/edits examined (**Rebuttal Figure 3B, C**). Next, a DNA-PK inhibitor (NU7441, different from the one used by Peterka et al.) was applied for inhibition of NHEJ in conjunction with the introduction of PEn. Notably, NU7441 could selectively improve the purity of RT-dependent edits by PEn (**Rebuttal Figure 3D, E**). These results demonstrate that inhibition of NHEJ leads to much higher levels of desirable editing at the expense of the imprecise edits with insertion of RTT sequences, provide validation for the latest work by Peterka et al. We have placed these results to Supplementary Fig. 1 (line 123-149).

Rebuttal Figure 3. The editing outcome of PE-nuclease and enhanced editing efficiency by inhibiting NHEJ repair pattern.

a. Example of PE-nuclease-induced 6*His insertion at *LSP1* site in HEK293T cells. Gray fill indicates the spacer region of pegRNA. Red arrows indicate the position of pegRNA-induced DSB. Blue fill indicates the insertion sequence. Yellow fill indicates the extra insertion sequence. The frequency of every genotype is showed in right. The statistic of genotype is showed in bottom.

b, c. Evaluation of PE-nuclease in HEK293T for targeted TAG (b) or 6*His (c) insertion. All RT-driven edits are consisting of desired edits and imperfected edits. Accurate edits represent desired edits. Values and error bars reflect the mean and s.d. of three biological replicates.

d. Prime editing outcomes of PE-nuclease in HEK293T cells treated with DMSO, 3, 6, 9 μ M NHEJ inhibitor. Editing for targeted TAG, 6*His and FLAG insertion in *FANCF*, *LSP1*, and *RUNX1*, respectively. Values and error bars reflect the mean and s.d. of three biological replicates.

e. The ratio of accurate edits relative to all RT-driven edits in (d). The darker the color the higher the percentage of accurate edits.

(II) Mechanistic insights on the desirable editing by uPEN:

We apologize for having not explored and discussed about the mechanisms underlying uPEN in the original manuscript.

The DNA-PK inhibitor-assisted PEn has established a proof-of-principle for refining PEn. To establish a more practical, protein-based strategy for PEn improvements, we adopted a previously established i53 module to inhibit 53BP1, a critical factor that limits DSB end resection for promotion of NHEJ. This led to the development of uPEn (Fig. 1 in the original and updated manuscript). Similar to the results of NU7441-assisted PEn above, i53 mainly acted via improving the purity of RT-driven edits by PEn to empower much higher levels of intended edits than PEn alone. We have revised the text, to further clarify the effects by i53 on the outcome of PEn editing (line 177-188). Moreover, we characterized the impacts by the length of homology region (HR) in the RTT on uPEn-dependent editing (Fig. 2 in the original and updated manuscript). The results demonstrated that the HR segment (with optimal effects reached at ~15-bp) within the reverse-transcribed 3' ssDNA overhang structure is essential for uPEn-associated precise DSB repair.

DSBs are repaired by the competing pathways of NHEJ and HDR, the latter of which is known to underlie Cas9/template-dependent precise genome editing. However, since effective HDR processes require operatable templates and at least 30-bp homology between the DSB end and the template, it is less likely that uPEn-dependent precise editing is mediated by HDR pathways. Nevertheless, for formal examination, we targeted HDR pathways by using inhibitors against Rad51 or Rad52 (see **Rebuttal Figure 4**). The results show neither the Rad51-dependent classic HDR pathway nor other Rad52-dependent pathways such as single-strand annealing and single-stranded template repair is involved in precise editing by uPEn (or by the less efficient PEn).

Another DSB repair mechanism that operates without the need of homologous templates is the microhomology-mediated end joining (MMEJ) pathway. Different from the NHEJ pathway, MMEJ-dependent repair obliges DSB end resection⁴, which may be potentiated by 53BP1 blockade⁵. In addition, MMEJ requires small-sized homology at the DSB ends, which is compatible with results of HR size-characterization above (Fig. 2). However, by using an inhibitor against PARP1, a key factor for canonical MMEJ, the accurate and unintended edits by uPEn or PEn were also not affected (see **Rebuttal Figure 5**). Despite the lack of effects by PARP1 inhibition, it remained plausible that a pathway partially resembling the MMEJ pathway might be involved in precise editing by uPEn/PEn (see working model in **Rebuttal Figure 6**). Given the sufficiency of uPEn/PEn to establish a 3' ssDNA overhang at the PAM-distal DSB end, the uPEn/i53-derepressed resection of the remaining PAM-proximal DSB end (likely in a PARP1-independent manner) could readily establish the other juxtaposed 3' ssDNA overhang. Consequently, specific base-pairing between the homologous regions on the pair of overhang structures would yield the key intermediate for the eventual installation of precise edits. Unlike the canonical MMEJ repair where resection leads to loss of genetic information, herein the uPEn-dependent reverse transcription of HR contributes to bypassing such an erroneous repair stage. The results and model are added to the manuscript (Supplementary Fig. 7-9, line 295-341).

This conceived mechanism for uPEn-mediated editing is also compatible with the observation that although uPEn effectively improved the purity of the RT-driven edits in comparison to PEn, it did not mitigate the induction of classical indels. It may be

reasoned that even with i53-stimulated resection, a “naïve” DSB that has not been modified a by RT-dependent upstream 3’ overhang would still prone to erroneous end-joining. The related texts further detailing the mechanistic model for uPEn are now added to the Discussions (line 551-583).

Rebuttal Figure 4. Editing outcomes of PEn and uPEn in HEK293T cells treated with Rad51 or Rad52 inhibitors.

a. The editing efficiency of targeted 34-bp insertion induced by PEn or uPEn at the *LSP1* sites in HEK293T cells. Cells were treated with indicated doses of Rad52 inhibitor (D-103) and Rad51 inhibitor (RI-1), respectively. Values and error bars reflect the means and standard deviation (s.d.) of three biological replicates.

b. The editing efficiency of targeted 24-bp insertion induced by PEn or uPEn at the *EMX1* sites in HEK293T cells. Cells were treated with indicated doses of Rad52 inhibitor (D-103) and Rad51 inhibitor (RI-1), respectively. Values and error bars reflect

the means and standard deviation (s.d.) of three biological replicates.

c. The editing efficiency of targeted 34-bp insertion induced by PEn or uPEn at the *FANCF* sites in HEK293T cells. Cells were treated with indicated doses of Rad52 inhibitor (D-103) and Rad51 inhibitor (RI-1), respectively. Values and error bars reflect the means and standard deviation (s.d.) of three biological replicates.

Rebuttal Figure 5. Editing outcomes of PEn and uPEn in HEK293T cells treated with PARP1 inhibitor.

a. The editing efficiencies by PEn for targeted 34-bp insertions at *FANCF*, *RUNX1*, *SEC61B* sites in HEK293T cells. At every site, cells were treated with indicated doses of PARP1 inhibitor (rucaparib). Values and error bars reflect the means and standard deviation (s.d.) of three biological replicates.

b. The editing efficiencies by uPEn for targeted 34-bp insertions at *FANCF*, *RUNX1*, *SEC61B* sites in HEK293T cells. At every site, cells were treated with indicated doses of PARP1 inhibitor (rucaparib). Values and error bars reflect the means and standard deviation (s.d.) of three biological replicates.

Rebuttal Figure 6. The potential mechanism(s) underlying PEn- and uPEn-dependent editing.

An illustration depicting the potential intermediates and repair mechanisms associated with PEn- and uPEn-dependent DSB is shown. The nuclease activity in PEn initially generates a blunt-ended DSB, with the upstream (PAM-distal) end readily undergo pegRNA-programmed reverse transcription to form a 3' overhang structure (upper part). If the DSB is repaired prior to the RT action, the NHEJ pathway would lead to classical indel formation (lower left). On the other hand, with the RT-generated overhang structure, the predominant NHEJ process would attempt to join such RT-driven overhang structure to the downstream, end-protected DSB terminus (PAM-proximal), causing imprecise edits by PEn (apparent insertion of RTT-derived sequences). Contrastingly, the i53 activity in uPEn would stimulate the resection of the downstream end of DSB, exposing another 3' overhang structure capable of specific base-pairing with the HR portion of the upstream overhang. This would yield the key intermediate for the eventual installation of precise edits.

(III) Control experiments supporting that the higher levels of desirable editing by uPEn is due to mechanistic improvements:

We thank the reviewer very much for point out other possibilities (e.g., intrinsic differences at levels of PE/Cas9 components or pegRNAs) that may contribute to the featured editing outcomes by uPEn. We therefore performed further control

experiments and more thorough data examinations to support our mechanistic view.

The i53-adopted uPE_n platform has shown marked improvements in desirable/unintended editing ratios over PE_n. To rule out that such improvements might be caused by hypothetical artifacts including the levels of PE or pegRNA expression, cells transfected with uPE_n3- or PE_n-pegRNA were harvested for expression analyses (a group of PE2max as an additional control). Little difference in the expression of PE protein and the corresponding pegRNA was observed among the three groups (Rebuttal Figure 7). These results are added to the manuscript (Supplementary Fig. 4F, line 216-221).

Rebuttal Figure 7. Comparisons of PE and pegRNA levels upon transfection with PE2max, PE_n and uPE_n.

HEK293T cells were transfected with PE_n, uPE_n3 and PE2max (as a control) together with the pegRNA targeting at the *UBE3A* site. The protein samples were analyzed by WB (top). The levels of pegRNA were examined by qPCR (bottom).

Our results also showed that uPE_n outperformed the conventional Cas9/template strategy for installing correct edits (see results in Rebuttal Figure 2). This most likely reflect the specific editing mechanisms by uPE_n (see Rebuttal Figure 6 and related discussions). It is worth pointing out that when comparing the two Cas9/template groups and the PE_n group, we found that they all featured ~70% (medians) unintended edits. Therefore, the similar indel levels between the Cas9/template groups and the PE_n group served as a control to show that the corresponding core editing components were likely to exhibit equivalent expression levels in the cells, despite the structural differences between Cas9/sgRNA and PE_n/pegRNA. Such analyses are detailed in the updated versions of manuscript (Supplementary Fig. 5, line 236-241).

3. Benefits of PE nucleases are unclear. Compared to the original Prime editing paper by Anzalone et al. from David Liu's lab, uPE_n only has marginally improved editing rates at the cost of significantly increased indels due to DSB activity instead of nickase. This appears to be a major drawback. Off-targets were a major concern original of Cas9 DSB induced editing. This was reduced by using a nickase variant for PE, but PE_n reintroduces this. Does uPE_n have reduced OT activity vs. PE or PE_n?

Bioinformatically determined OT sites are often insufficient, do the authors have work showing that biologically determined OT sites are less so affected?

[Authors' Responses]

We thank the reviewer for this series of questions. We have attempted to address the first part of the question in our earlier responses (see our response to point #1 above). In regards to the off-target-related issues, we have performed further experiments and also carefully revised the manuscript for clarifications.

Off-target analyses for uPEn:

We concur with the reviewer that PEn/uPEn-induced dsDNA cleavage has introduced off-target-related safety concerns, as compared to the canonical, nickase-based PE. For examination of off-target effects by PEn and uPEn, selected analyses of established off-target sites had been presented in the original manuscript (Supplementary Fig. 5, currently at Supplementary Fig. 15). These sites are indeed the proven, higher-ranked off-target sites for given sgRNAs previously determined by GUIDE-seq⁶. We apologize for having not made sufficient clarification in the original text. The results showed uPEn and PEn induced mostly similar levels of off-targeting at these sites.

For a broader view of off-targeting by uPEn and PEn, and for comparison with canonical PE, we further conducted whole-genome resequencing to systemically profile off-target effects by PE5max, PEn and uPEn when the *FANCF* site was targeted for insertions. Cells transfected with EGFP-expressing plasmid were used as control for the cells subjected to editing. The whole-genome sequencing (WGA) was performed using genomic DNA from different groups of cells [sorted on EGFP after transfection] (see Rebuttal Figure 7A, B). Potential guide RNA-dependent off-target sites with up to 5 mismatches from the target sequences were selected for further analyses (see Rebuttal Figure 7C). Importantly, via comparing the reads from the control and experiment groups, only one off-target edit (at a site with 2-nt mismatches, same as the corresponding “OT1” site in Supplementary Fig. 15) was identified in both the PEn and uPEn groups, but not in the PE5max group. Therefore, such WGS data confirmed elevated levels of off-targeting by PEn or uPEn over PE5max, while suggesting no overt differences of off-targeting properties between the two PEn platforms. These results are now added to the updated versions of manuscript (Supplementary Fig. 16, with related texts at (line 473-492)).

Such increased off-target risks by uPEn were expected, due to the re-introduction of Cas9-nuclease, as we had originally stated in the related texts of the Results and Discussions. We added texts in the Introduction (line 107-111) and in the Results (line 464-465; 488-492) to further convey the information. On the other hand, it may be envisioned that the adoption of higher-fidelity Cas9 variant may help to mitigate uPEn-associated off-targeting (as we had originally mentioned in the Discussions). Therefore, we believe that the significantly higher potencies by uPEn than the canonical PE to install desirable edits has suggested uPEn as a useful platform to complement the existing PE tools, especially when higher levels of productive edits representing a

priority. Such comments are also added to the Results in the updated version of the manuscript (line 495-500).

A

Sample	Sequenced reads	Mapping rate	Duplication	Mean depth
Untransfected	604,744,838	95.71%	17.26%	20.34X
Control	564,646,676	98.98%	14.82%	21.67X
PE5max	569,574,624	98.60%	14.83%	20.72X
PEn	577,800,824	98.61%	16.11%	21.38X
uPEn	561,389,458	98.72%	17.71%	21.09X

B

	Control	PE5max	PEn	uPEn
C:G > T:A	19,123	18,453	17,998	17,209
C:G > G:C	4,712	4,453	4,384	4,202
C:G > A:T	5,691	5,553	5,420	5,242
A:T > G:C	16,538	15,901	15,450	14,749
A:T > C:G	4,656	4,738	4,591	4,461
A:T > T:A	4,275	4,142	3,958	3,824
Indels	63,021	61,456	61,519	63,158

C

Mismatch	Predicted	Control	PE5max	PEn	uPEn
1	0	0	0	0	0
2	1	0	0	1	1
3	38	0	0	0	0
4	438	0	0	0	0
5	2,954	0	0	0	0

Rebuttal Figure 7. Whole genome sequencing analyses of off-target effects associated with PE5max, PEn and uPEn.

- Whole-genome sequencing sample information. HEK293T cells were respectively transfected with EGFP (control); PE5max, PEn and uPEn. The *FANCF* site was targeted in cells transfected with various PE platforms.
- Summary of total unique variants and indels detected by WGS in control, PE5max, PEn, and uPEn samples.
- Summary of off-target analysis. Potential off-target sites with up to 5-nt mismatches from the spacer of the targeting pegRNA were predicted by Cas-OFFinder (requiring NRG as PAM).

4. Side effects of i53 are not properly monitored. The original suggestion that NHEJ inhibition by small molecule DNA-PK (AZD7648) has non-specific effects and may present toxicity effects is a valid concern. However, the authors do not show whether their i53 ubiquitin variant is toxic. Furthermore, the results suggest that this ubiquitin variant works best with cell-wide administration (i.e. uPEn variant 1 and 3). Is overexpression of this ubiquitin variant in the nucleus inhibiting NHEJ or

sequestering 53BP1 in a non-reversible manner? Is an increase in general DNA damage/toxicity in treated cells? Is there an increased probability of neoplastic events?

[Authors' Responses]

We thank the reviewer for this series of questions, and apologize for the lack of direct testing and discussions of potential toxicity of i53/uPEn in the original manuscript.

Therefore, we evaluated the potential cellular toxicity associated with i53 expression alone or \pm PEn components, at the same time point when the editing performances were determined [72 h post-transfection of HEK293T cells] (see **Rebuttal Fig. 8A**). The expression of i53 alone at different transfected doses did not affect the cell viability. Furthermore, no differences in viability were observed among PE2max, PEn and uPEn3 groups. The transient expression of i53 was shown by WB of Flag-tagged i53 (see **Rebuttal Fig. 8B**). These results validated the minimal toxicity associated with transient transfections of i53 and of uPEn. This part is added to the revised manuscript (Supplementary Fig. 6, line 258-268).

The reviewer's comments also reminded us that it would be important to inform the readers regarding the specific activity of the i53-Ubv tool that we have adopted. The i53 module is a previously engineered ubiquitin variant that is uncovered by the characteristics of binding to the Tudor domain of 53BP1 with high affinity⁷. Such binding would sequester 53BP1 from recruitment to DSBs. This Ubv also harbors mutation of the terminal diglycine module, which eliminates its possible "non-specific" function as a substrate for covalent conjugations. The additionally featured I44A mutation in i53 further diminished its potential activity to mediate other ubiquitin-like associations⁷. Some modifications are made to the Results in the revised manuscript to underscore the functional specificities of the Ubv-s tested (line 156-159).

In addition, we have noted that a number of reports on the application of i53 or similar modules in genome editing have investigated the associated safety profiles. The results have shown that short-term co-administrations of i53 (or other modules of 53BP1 inhibition) and the CRISPR/Cas9 apparatus to the cells did not reveal increased levels of genomic damages, translocations events, or other adverse effects even in models of edited HSPCs monitored after in vivo reconstitution. Although further safety evaluations are surely warranted, we believe that these lines of evidence support the feasibility of adopting i53 in transient genome editing applications. Such accounts have now been added to the Discussions of the revised manuscript (line 584-595).

Rebuttal Figure 8. Examination of cell viability in response to the transfected i53 or uPEn.

a. Measurements on the viability of HEK293T cells transfected with PE2max, PEu, uPEu or different amounts of i53 in 96-well plates. The assays were carried out using the luminescence detection kit. Values and error bars reflect the means and standard deviation (s.d.) of three biological replicates. *P* values were calculated by a two-tailed Student's *t*-test.

b. Western blot analysis of Flag-i53 expression in HEK293T cells at indicated time points after transfection.

Reviewer #2 (Remarks to the Author):

The ability of CRISPR prime editing to efficiently introduce precise editing including small indels and substitutions has made this technology to become the next generation gene editing technique. Prime editing is originally derived from the fusion of Cas9-nickase and reverse-transcriptase. As of now, several strategies and novel improvements to use this technology have been developed. PE2 strategy relies on single gRNA and single-strand break therefore avoiding DNA double strand break. However, PE2 efficiency is low. Additional gRNA targeting the opposite strand (second nick) enhances the prime editing efficiency, a technique dubbed as PE3 which basically creates a staggered-DSB similar to a dual-nickase approach. Optimization of the codon usage and NLS has created PEMax. PE3 with this optimization is called PE3-Max and addition of MMR inhibitor which is called PE5-Max was claimed to improve prime editing.

To bypass the need of the second-nick gRNA for enhanced prime editing, Adikusuma et al generated prime editing nuclease. Although prime editing initiation was robust, it suffered from unwanted prime editing outcomes whereas the transcribed homology undergoes non-homologous end joining (NHEJ) instead of strand annealing.

In this study Li et al cleverly used NHEJ inhibitor called i53 discovered by Canny et al that can inhibit the key player of NHEJ, the 53BP1. They found that this strategy could improve accurate prime editing as opposed to unwanted template insertions as mostly seen in the standard nuclease prime editor.

This study is interesting and the finding is important in the field of CRISPR gene editing. This finding will contribute significantly to improve and simplify prime editing technique.

[Authors' Responses]

We thank the reviewer very much for his/her overall positive appraisal for our study. We believe that uPE platform developed in the present study represents a significant upgrade from the original PE platform, and enriches the PE toolkit to suit different editing needs. Despite the caveats of inducing certain levels of collateral Cas9-dependent indels and off-targets, uPE demonstrates high potencies for intended edits. During the revision of our manuscript, we further extended our comparisons of uPE and the canonical PE platforms (see Rebuttal Figure 1, 2), demonstrating the superior potencies by uPE to the canonical PE platforms to install a variety of edits. These results are added to the revised manuscript (Supplementary Fig. 5 and 14, line 222-257; 447-460). Therefore, our work establishes an up-graded PE platform to complement the existing PE tools, especially when higher levels of productive edits representing a priority.

However, there are some concerns that I would like the authors to address:

The NHEJ inhibitor factors used in this study was based on Canny et al (Nat Biotech 2019). Canny et al tested several candidates and found UbvG08(I44A) as the best 53BP1 inhibitor and dubbed this factor as "i53". However, authors did not mention i53 at all.

[Authors' Responses]

We appreciate this important reminder. We now describe the previous specification of "i53" in relation to our screening results (line 188-190). The term i53 is subsequently used throughout the manuscript.

Line 77: Authors stated the uncertainty of specificity/toxicity of small molecule inhibitors. However, authors could not provide evidence the harmless of overexpression of the UbvG08 factors.

[Authors' Responses]

We thank the reviewer for this important question that was also raised by the other reviewer. Therefore, we evaluated the potential cellular toxicity associated with i53

expression alone or \pm PEn components, at the same time point when the editing performances were determined [72 h post-transfection of HEK293T cells] (see **Rebuttal Fig. 8A**). The expression of i53 alone at different transfected doses did not affect the cell viability. Furthermore, no differences in viability were observed among PE2max, PEn and uPEn3 groups. The transient expression of i53 was shown by WB of Flag-tagged i53 (see **Rebuttal Fig. 8B**). These results validated the minimal toxicity associated with transient transfections of i53 and of uPEn. This part added to the revised manuscript (Supplementary Fig. 6, line 258-268).

In addition, we have noted that a number of reports on the application of i53 or similar modules in genome editing have investigated the associated safety profiles. The results have shown that short-term co-administrations of i53 (or other modules of 53BP1 inhibition) and the CRISPR/Cas9 apparatus to the cells did not reveal increased levels of genomic damages, translocations events, or other adverse effects even in models of edited HSPCs monitored after in vivo reconstitution. Although further safety evaluations are surely warranted, we believe that these lines of evidence support the feasibility of adopting i53 in transient genome editing applications. Such accounts have now been added to the Discussions of the revised manuscript (line 584-595).

Fig 1E: Although I understand this graph, other readers might get confused with the 'indel' bars.

[Authors' Responses]

We thank the reviewer for pointing out such a cause of potential confusion. Indeed, the unintended editing outcome of PEn include both classical indels and imprecise RT-driven edits. Therefore, we used the term "Unintended edits" as a sum of both compartments. Such specification is now added to the revised manuscript (line 201-205).

Introduction: It is better if the authors remind the readers regarding PE2, PE3 and PE5 systems as well as reminding that the PE nuclease uses only a single spacer instead of dual-guides as in PE3 or PE5.

[Authors' Responses]

We appreciate this suggestion. We now extend our introduction of PE to the major iterations (line 65-70). In addition, the convenience by PEn to use a single guide RNA is added to the text (line 79-80; and again in Discussions, line 517-518).

Line 120: Adikusuma et al (NAR 2021) observed high prime editing event but mostly resulted in unwanted template insertions (from PEn). It is unclear in this study whether improved accurate prime editing efficiency is due to the reduction of

unwanted templated insertions that are converted to correct prime editing. If it is the case, authors need to make a bold statement and make it clear in the figure (Fig 1) by including the data showing the frequency of the unwanted template insertions.

[Authors' Responses]

We thank the reviewer very much for these comments. The abundance of RTT sequence-containing, but imprecise edits is indeed a prominent feature of PEn. The improved activities by uPEn are indeed attributed to the “conversion” of such RT-dependent imprecise edits into desirable edits. We have made efforts to elaborate and stress such a point throughout the manuscript.

In the original manuscript, we have attempted to use the ratio of “Accurate edits/All RT-driven edits” to compare the purity of RT-driven edits by different versions of PEn (original Supplementary Fig. 1, current Supplementary Fig. 2). In addition, to further illustrate the reduction of such imprecisely repaired byproducts by 53BP1 inhibition, we also presented the results from the above experiments in the form of relative “Imprecisely repaired edits” levels within “all RT-driven edits” in the control, Ub (wt), UbvG08 and G08 (I44A) groups (see Rebuttal Figure 9A). Relative levels of imprecise RT-driven edits normalized to all sequencing reads in these groups of PEn samples exhibited a very similar trend (see Rebuttal Figure 9B). Such an additional form of data presentation has been added to the revised manuscript (Supplementary Figure 3, line 184-188).

Rebuttal Figure 9. 53BP1-targeting ubiquitin variants decreased PEn-associated

imprecise edits to improve the purities within RT-driven edits.

a. The ratios of imprecisely repaired edits to all RT-driven edits at *LSP1*, *SEC61B* and *RUNX1* sites in HEK293T cells co-introduced with PEn and the WT Ub, G08 or G08 (I44A) ubiquitin variants.

b. The frequencies (among all reads) of imprecisely repaired edits at *LSP1*, *SEC61B* and *RUNX1* sites in HEK293T cells co-introduced with PEn and the WT Ub, G08 or G08 (I44A) ubiquitin variants. Values and error bars reflect the means and s.d. of three biological replicates.

Line 144 and Figure 1E: It is important to include the frequency of unwanted template insertions to emphasize that increased PE was due to the conversion of unwanted template insertions to correct PE. This data is crucial and I expect this important observation is also seen across multiple cell lines.

[Authors' Responses]

We thank the reviewer again for stressing the need to better explain the enhanced efficiencies by 53BP1 inhibition. Although such information has been included in the original manuscript (original Supplementary Fig. 2, current Supplementary Fig. 4), we sincerely apologize for having not conveyed the point with sufficient clarity. We have made efforts to highlight this point in the corresponding text (line 205-209).

The above results are from editing of HEK293T cells. With the available editing data also from U2OS cells (original Supplementary Fig. 3), we now further quantitated the ratios of “Accurate edits/All RT-driven edits” (see Rebuttal Figure 10B). The results showed that uPEn also led to high-purity desirable products (up to 80%, and at least 60%) within RT-dependent edits in U2OS cells. The results for such analyses have been added to the revised manuscript (Supplementary Fig. 10B, line 364-367).

Rebuttal Figure 10. Targeted sequence insertion, deletion and recoding by uPEn system in U2OS cells.

a. Targeted insertion, deletion and replacement edits with uPEn at multiple sites in U2OS cells. Values and error bars reflect the mean and s.d. of three biological replicates.

b. The ratio of uPEn-induced accurate edits relative to all RT-driven edits at the targeted

sites in (a).

Line 130: given no experimental comparison was performed, I suggest moving this statement to Discussion.

[Authors' Responses]

We thank the reviewer for this suggestion. We have placed this comment into Discussion of the revised manuscript (line 534-536).

All data (including supplementary) should show individual values by using scattered plot bar.

[Authors' Responses]

We apologize for having not displayed all data points in our graphs. We have updated all figure according to this requirement.

Figure 1B is a representative of 1 of 3 sites performed by authors. It would be better to see all 3 sites in one graph. To save some space I would suggest the graph to contain only control, Ub (WT), G08 and G08(I44A) and move the remaining to the supplementary data.

[Authors' Responses]

We thank the reviewer for this suggestion. Figure 1B has now been re-organized to include information for all three sites. The results for the remaining Ubv-s can now be found in Supplementary Figure 2.

Line 157: It is too immature to draw strong conclusion of the optimal length of the homology arms based on this experiment given that only insertions of >24 bp were performed. Therefore I would suggest toning down the sub-heading e.g. "optimizing the size of...." to indicate that this optimal size might be relevant only for this study rather than for general use.

[Authors' Responses]

We thank the reviewer for this important point. Now we have revised our manuscript to prevent overstatements. The sub-heading has been changed into "Impacts by the size of the homologous region in pegRNAs on uPEn-dependent editing" (line 270-271).

Figure 2D: The low efficiency in PAH and FANCF could be caused by longer inserts? Can be mentioned in the text.

[Authors' Responses]

We thank the reviewer for highlighting this observation. Larger changes are intrinsically more difficult to install by PE platforms⁸. We have now added such reasoning in the revised manuscript (line 292-294).

Line 172: chose instead of choose.

[Authors' Responses]

We apologize for the mistake. The correction is made (line 287).

Line 181 & 211: Worth mentioning that uPEn-3 system was used.

[Authors' Responses]

We thank the reviewer for pointing out this clarity issue. Despite our original statement after describing the results of Fig. 1E, F (“In the ensuing experiments, we further chose to mainly focus on the uPEn3 system due to its more compact configuration.”), some more emphases elsewhere might have been needed. To this end, we now specify the use of term “uPEn” as “uPEn3” in the Figure legends to Fig. 2, 3 and 4 (line 940, 955, 969).

Line 184: Six genomic loci however there are only 3 apparently.

[Authors' Responses]

We apologize for the mistake. We have corrected this with the accurate information (line 348-349).

The uPEn system was not tested in hard-to-prime-edit cells such as HeLa.

[Authors' Responses]

We thank the reviewer for this important suggestion. It is known that PE platforms operate inefficiently in certain cell types, including the commonly used HeLa cells. Therefore, given the comparisons of uPEn and PE3/5max activities in HEK293T and U2OS cells, we subsequently carried out similar experiments in HeLa cells (see **Rebuttal Figure 11A, B**). The base conversion efficiencies (at 6 sites) by uPEn and PE5max in HeLa cells were examined. Notably, uPEn empowered precise base conversions with a median level of 14% efficiency, compared to a corresponding level of 2% by PE5max. The uPEn achieved more than 20% (up to 40%) efficiency at 3 out

of 6 sites, whereas PE5max led to more than 10% (15% at the maximum) precise edits in only 2 out of 6 sites. While the precise base conversion rates by uPEn remained suboptimal in HeLa cells compared to other cell types, they were significantly higher than those by PE5max in this cell type. The nevertheless substandard patterns of accurate/unintended editing ratios by uPEn in HeLa cells (<1 for 5 out of 6 sites, unlike in HEK293T and U2OS cells) point to the existence of other limiting factors for productive applications of PE_n in this cell type. These results are added to the revised manuscript (Supplementary Fig. 12, line 404-416).

Rebuttal Figure 11. Comparisons of uPE_n and PE5max for installing base conversions in HeLa cells.

- Comparison of base conversion efficiencies and the levels of unintended edits by PE5max and uPE_n at six endogenous sites in HeLa cells. Values and error bars reflect the mean and s.d. of three biological replicates.
- Statistical analysis of desirable base conversion efficiencies of PE5max and uPE_n in HeLa cells. Boxes represent 25th–75th percentile (line at the median). n = 6 for each group. *P* values were calculated by a two-tailed Student's *t*-test.

Line 203-238 and Material & Method: In this study, authors generated the PE_n and uPE_n using the backbone of PE-Max which uses optimized codon and has extra NLS. However, when comparing the uPE_n with PE3 and PE5, authors used PE2 construct for the PE3 and PE5 systems that doesn't have optimized codon and extra NLS. To me this is the major flaw of this study. To enable fair comparison, authors should use PE2-Max construct for the comparison.

[Authors' Responses]

We thank the reviewer very much for underscoring the need of parallel comparisons. Indeed, similar to our construction of the PE_n and uPE_n platforms (as we have originally specified), the PE2, PE3 and PE5 platforms used in the study were also based on the PE_{max} backbone. Now we have used the suffix of “max” in denoting various versions of canonical PE platforms throughout the text.

Methods: the amount of lipofectamine was not mentioned.

Methods: uPEn transfection ◊ 900 ng uPEn plasmid + 300 ng pegRNA plasmid
PE3 transfection ◊ 900 ng PE2 + 300 ng pegRNA + 100 ng nick gRNA plasmid
PE5 transfection ◊ 900 ng PE2 + 300 ng pegRNA + 100 ng nick gRNA + 450 ng hMLH1dn plasmid

[Authors' Responses]

We apologize for having not provided such information. This has been corrected (line 618-619).

In addition to using PE2-Max construct, I would like to point out the possibility of inefficient transfection of the extra plasmids (nick gRNA + hMLH1dn plasmids). It would be a more fair comparison if the nick gRNA is expressed in the same plasmid as the pegRNA.

For comparison with PE5, plasmid PE-Max-P2A-hMLH1dn <https://www.addgene.org/174828/> should be used. This will rule out the possibility that inefficiency of PE3/PE5 is caused by inefficient transfection due to the presence of extra plasmids.

Comparison with PE3max and PE5max should also perform editing the same as previously targeted by David Liu.

[Authors' Responses]

We thank the reviewer very much for this series of questions. Indeed, the control of plasmid numbers among comparison groups needed to be considered. It is also important to include applications of previously reported edits. We therefore performed additional experiments in regards to both considerations above.

The delivery of PE3max and PE5max would require greater numbers of plasmids than uPEn. Therefore, additional control experiments were carried out. For simplification of the transfection step for PE5max, a singular construct was previously established by connecting the PEmax and the MLH1dn modules via P2A⁹. We used the abbreviation of "PE5maxP" (with the "P" denoting P2A) to specify the application of this construct together with pegRNA and nick-sgRNA in transfection. Furthermore, we also established dual-guide RNA-expressing constructs by placing the pegRNA and nick-sgRNA in a same plasmid (respectively under a U6 promoter). Co-transfection of PE5maxP with such a construct would be specified as "PE5maxP-2U6". In this configuration, the PE5max platform could be transfected via 2 plasmids, a format similar to the delivery of uPEn. Next, focusing on three base-conversional edits previously analyzed by others⁹, we compared the effects by PE3max, PE5maxP, PE5maxP-2U6 and uPEn in HEK293T cells (see Rebuttal Figure 12). Across these sites, PE5maxP led to only slight increases in efficiencies over PE3max, generally consistent with the previous report⁹. In addition, the combination of guide RNAs in a same plasmid (PE5maxP-2U6) did not result in higher editing levels than the levels by

PE5maxP, excluding the combination of plasmids as a burden for canonical PE. Importantly, the uPE_n drove markedly higher levels of precise edits than PE5maxP-2U6 (and all other groups) in two out of three sites (Rebuttal Figure 12). At the remaining site (within *CXCR4* locus), uPE_n enabled an equivalent level of editing in comparison to that by PE5maxP-2U6. Interestingly, we noted that PE3/5 activities were already robust (~60%) at this site. Here, the overall higher activities of uPE_n compared to various PEmax platforms or configuration-simplified versions, especially in installing less editable base conversions (see sites of *CDLK5* and *IL2RB*), further corroborated our earlier results in HEK293T, U2OS and HeLa cells. These results have been added to the revised manuscript (Supplementary Fig. 13, line 417-443).

Rebuttal Figure 12. Comparisons among uPE_n, PE3max and configuration-simplified PE5max for base conversions at previously tested sites.

Comparison of base conversion efficiencies and indels induced by PE3max, PE5maxP, PE5maxP-2U6 and uPE_n at three reported sites in HEK293T cells. PE5maxP: a construct with PEmax and the MLH1dn modules connected via P2A. PE5maxP-2U6: PE5maxP transfected together with a plasmid containing both U6-driven pegRNA and nick-sgRNA cassettes. Values and error bars reflect the mean and s.d. of three biological replicates.

Generated plasmids are expected to be available to the community through Addgene.

[Authors' Responses]

We thank the reviewer for the suggestion. We are in the process of banking the plasmids.

In summary, this manuscript requires major reconstruction and considerably more experimental data to confirm the findings and draw reliable conclusions.

[Authors' Responses]

We thank the reviewer very much for the constructive comments and suggestions. In the revised manuscript, we have addressed all the points raised. We believe that the manuscript is now significantly improved.

- 1 Nambiar, T. S., Baudrier, L., Billon, P. & Ciccia, A. CRISPR-based genome editing through
the lens of DNA repair. *Mol Cell* **82**, 348-388, doi:10.1016/j.molcel.2021.12.026 (2022).
- 2 Adikusuma, F. *et al.* Optimized nickase- and nuclease-based prime editing in human
and mouse cells. *Nucleic Acids Res* **49**, 10785-10795, doi:10.1093/nar/gkab792 (2021).
- 3 Peterka, M. *et al.* Harnessing DSB repair to promote efficient homology-dependent and
-independent prime editing. *Nat Commun* **13**, 1240, doi:10.1038/s41467-022-28771-1
(2022).
- 4 Deriano, L. & Roth, D. B. Modernizing the nonhomologous end-joining repertoire:
alternative and classical NHEJ share the stage. *Annu Rev Genet* **47**, 433-455,
doi:10.1146/annurev-genet-110711-155540 (2013).
- 5 Zimmermann, M. & de Lange, T. 53BP1: pro choice in DNA repair. *Trends Cell Biol* **24**,
108-117, doi:10.1016/j.tcb.2013.09.003 (2014).
- 6 Tsai, S. Q. *et al.* GUIDE-seq enables genome-wide profiling of off-target cleavage by
CRISPR-Cas nucleases. *Nat Biotechnol* **33**, 187-197, doi:10.1038/nbt.3117 (2015).
- 7 Canny, M. D. *et al.* Inhibition of 53BP1 favors homology-dependent DNA repair and
increases CRISPR-Cas9 genome-editing efficiency. *Nat Biotechnol* **36**, 95-102,
doi:10.1038/nbt.4021 (2018).
- 8 Anzalone, A. V. *et al.* Search-and-replace genome editing without double-strand breaks
or donor DNA. *Nature* **576**, 149-+, doi:10.1038/s41586-019-1711-4 (2019).
- 9 Chen, P. J. *et al.* Enhanced prime editing systems by manipulating cellular determinants
of editing outcomes. *Cell* **184**, 5635-5652 e5629, doi:10.1016/j.cell.2021.09.018 (2021).

Reviewers' Comments:

Reviewer #1:

Remarks to the Author:

Congratulations to the authors on their new and substantially improved manuscript. Please see my original major concerns below and the extent to which I consider these concerns resolved. I would be happy to see a version of this manuscript in print.

Major concern 1: Comparisons made within figures are incomplete. – Resolved.

1. I find the data in Figure 3 and Figure S5 to be compelling.
2. I do wonder what would happen if PE2 was combined with i53, but do not suggest that the authors test this.

Major concern 2: There is little mechanistic insight. – Partially resolved.

1. The authors expand their discussion of mechanism and I find that this improves the manuscript. Their favored model seems to be that RT-mediated insertion in PEn is constant but 53BP1-mediated protection of the end of the DSB distal to RT activity can be inhibited by i53. This sets up a dynamic in which the RT-encoded modifications can anneal to the resected sequence distal to the DSB distal to RT activity. This annealing activity is what drives the increase in desired edit and occasional decrease in overall RT-mediated insertion.
2. The authors attempt some drug inhibition experiments to test this model. I find these to be unconvincing because the authors rely on negative (editing) data without showing that the treatments themselves work. Figure S1 is an exception. The authors do not show that their inhibition works, but as there is a dose-dependent effect on editing, I find the data compelling. Figures S7 and S8 show no changes in editing at any tested doses. Absent some orthogonal validation of inhibition effectiveness, I am unwilling to interpret this data. From my perspective, the authors can just remove these figures and discussion from the text. If the data stays in the manuscript, I suggest that the authors validate their drug treatment with orthogonal assays and/or repeat with knockdown/knockout experiments.
3. One unexplored approach to build mechanistic understanding of this system is to look within the amplicon sequencing reads for evidence of the mechanism. Do the authors see evidence of more resection with i53 treatment? Is this resection asymmetric around the DSB, as their model predicts? I attempted to look at the sequencing reads posted for this manuscript but was unable to find enough description about samples to begin analysis. To be clear, I am not suggesting that the authors do additional experiments here, I am suggesting that they might be able to process existing data to support or refute models discussed in the manuscript. We have had good luck using CRISPResso2 as a preliminary analysis tool and then developing more robust analyses when appropriate. Evidence that i53 influences outcomes at single loci would add a lot of mechanism to the manuscript and reject the possibility that i53 is just globally scrambling DNA repair mechanisms.
4. I would like to thank the authors for uploading their sequencing data to a central repository and encourage them to better define data/analyses. Improved metadata on SRA would allow re-analysis of amplicon sequencing data. The authors should also include a discussion of their amplicon sequencing approach in the materials and methods section.

Major concern 3: Benefits of PE nucleases are unclear. – Resolved.

1. The authors have convinced me that this is an interesting activity and that this approach may be useful for applications in which total frequency of desired edits is the primary goal.

Major concern 4: Side effects of i53 are not properly monitored. – Resolved.

- a. The authors conclusively address this.

Reviewer #2:

Remarks to the Author:

All issues have been sufficiently addressed.

Suggestion: please consider moving supplementary figures 11-14 to the main figures.

MS title: Development of a versatile nuclease prime editor with upgraded precision, uPEn

Author: Li, et al.,

MS number: NCOMMS-22-23029

Responses to the reviewer's comments:

Reviewer #1

Congratulations to the authors on their new and substantially improved manuscript. Please see my original major concerns below and the extent to which I consider these concerns resolved. I would be happy to see a version of this manuscript in print.

Major concern 1: Comparisons made within figures are incomplete. – Resolved.

1. I find the data in Figure 3 and Figure S5 to be compelling.
2. I do wonder what would happen if PE2 was combined with i53, but do not suggest that the authors test this.

[Authors' responses]

We thank the reviewer very much for pointing out the improvements by our current manuscript in comparison to the initial submission.

We also appreciate the reviewer's question on whether i53 may also affect PE2. The enhanced performance by uPEn (over PEn) is consistent with an established role of 53BP1 in regulating DSB repair (but not the nick repair). Although the effect by i53 on the nickase-based PE2 awaits to be formally addressed, previous screening experiments in K562 and HeLa cells did not suggest a regulatory effect of 53BP1 on PE2/3 editing¹. We placed such information to the Discussions (line 607-610).

Major concern 2: There is little mechanistic insight. – Partially resolved.

1. The authors expand their discussion of mechanism and I find that this improves the manuscript. Their favored model seems to be that RT-mediated insertion in PEn is constant but 53BP1-mediated protection of the end of the DSB distal to RT activity can be inhibited by i53. This sets up a dynamic in which the RT-encoded modifications can anneal to the resected sequence distal to the DSB distal to RT activity. This annealing activity is what drives the increase in desired edit and occasional decrease in overall RT-mediated insertion.
2. The authors attempt some drug inhibition experiments to test this model. I find these to be unconvincing because the authors rely on negative (editing) data without showing that the treatments themselves work. Figure S1 is an exception. The authors do not show that their inhibition works, but as there is a dose-dependent effect on editing, I find the data compelling. Figures S7 and S8 show no changes in editing at any tested doses. Absent some orthogonal validation of inhibition effectiveness, I am unwilling to interpret this data. From my perspective, the authors can just remove these figures and discussion from the text. If the data stays in the manuscript, I

suggest that the authors validate their drug treatment with orthogonal assays and/or repeat with knockdown/knockout experiments.

[Authors' responses]

We thank the reviewer very much for these comments. In our hands, the several inhibitors against Rad51, Rad52, and PARP1 did not affect the editing efficiencies/purities by PEn/uPEn. However, we concur with the reviewer that when presenting such negative effects of pharmacological inhibitors, it would be best to include “orthogonal” controls for their efficacies. Therefore, we followed the reviewer’s suggestions to remove the above-mentioned inhibitor data. It may be anticipated that substantial experimental efforts would be required in the future to establish the detailed mechanisms underlying uPEn. Some transition sentences are added prior to our proposal of a working model (line 308-315).

3. One unexplored approach to build mechanistic understanding of this system is to look within the amplicon sequencing reads for evidence of the mechanism. Do the authors see evidence of more resection with i53 treatment? Is this resection asymmetric around the DSB, as their model predicts? I attempted to look at the sequencing reads posted for this manuscript but was unable to find enough description about samples to begin analysis. To be clear, I am not suggesting that the authors do additional experiments here, I am suggesting that they might be able to process existing data to support or refute models discussed in the manuscript. We have had good luck using CRISPResso2 as a preliminary analysis tool and then developing more robust analyses when appropriate. Evidence that i53 influences outcomes at single loci would add a lot of mechanism to the manuscript and reject the possibility that i53 is just globally scrambling DNA repair mechanisms.

[Authors' responses]

We thank the reviewer very much for these suggestions. Since 53BP1 promotes NHEJ via inhibition of DSB end resection, it is conceivable that the i53-stimulated resection of the downstream DSB end would lead to its base-pairing with the RT-dependent upstream 3'-overhang to drive the precise repair (see illustration in the current Supplementary Fig. 7, originally in Supplementary Fig. 9). For an initial exploration into such a proposed mechanistic model, we now have followed the reviewer’s helpful suggestion to re-examine the available sequencing reads.

Here, to seek evidence that could possibly report i53-mediated increase of end resection at uPEn-targeted sites, we closely examined the patterns of PEn- and uPEn-dependent alleles using the CRISPResso2 program. The results showed that the PEn/uPEn-dependent alleles with direct indels or with RT-dependent edits are mutually exclusive (current Supplementary Fig. 8). The RT-dependent, imprecise edits by PEn harbor indels distal from the guide RNA target. Most notably, uPEn was associated with effective mitigation of all PEn-enabled distal indels (Supplementary Fig. 8A, B), while it led to only moderately changed patterns of direct indels in comparison to PEn (Supplementary Fig. 8C, D). The outputs from the “sequence

alignment viewer” further elucidate that the great majority of the distal indels contained the intact 18-bp insertion, in conjunction with variably sized HR fragments (< 20-bp) apparently joined directly to the unaltered, downstream blunt end of DSB (Supplementary Fig. 8E). In contrast, with the same low-frequency cutoff, the uPEEn group demonstrated much fewer numbers of such directly-joined repair products, with the remaining ones featuring greatly reduced levels (Supplementary Fig. 8F). These particular changes in the patterns of PEEn/uPEEn-edited alleles are consistent with a model where the uPEEn (i53)-promoted resection of the downstream DNA end would create a complement, downstream overhang to correctly align the RT-dependent 3'-overhang, which would promptly empower a precise repair. These new results are presented in Supplementary Fig. 8 (line 326-351).

To examine the rates of DSB end resection from another perspective, we took advantage of our NGS data from a PEEn experiments (\pm NU7441) using a specific, HR-free version of pegRNAs (Supplementary Fig. 9). Owing to the lack of extended end-homology, the performances of such an atypical PEEn was not enhanced by DNA-PK inhibitor NU7441 (results added in Supplementary Fig. 1F, G [line 149-157]), unlike the situation in standard PEEn. On the other hand, we reasoned that such imprecise stitching of the RT-dependent 3'-overhang and the downstream DSB end may result in certain “genetic scar” to independently report the extent of DNA end resection. Indeed, our close examinations of the editing products (at *SEC61B* site) led to the identification of a deletional allele apparently resulting from microhomology-mediated end joining (MMEJ) of RT-driven overhang with the downstream DNA end (Supplementary Fig. 9C, D). Importantly, we found that NU7441 treatment significantly increased the levels of such an MMEJ allele (Supplementary Fig. 9E). Similar results were obtained from analyses of a different *FANCF* site (Supplementary Fig. 9F-J). As MMEJ is known to be initiated by DSB end resection², these results provided additional independent evidence for the increases of DSB end resection upon the combined actions of PEEn and NHEJ inhibition. These new results are included in current Supplementary Fig. 9 (line 352-377).

4. I would like to thank the authors for uploading their sequencing data to a central repository and encourage them to better define data/analyses. Improved metadata on SRA would allow re-analysis of amplicon sequencing data. The authors should also include a discussion of their amplicon sequencing approach in the materials and methods section.

[Authors' responses]

We thank the reviewer for suggesting us to provide more information regarding the experiments and samples related to the data uploaded to SRA. An excel sheet with such information (named “SRA”) has been included in Supplementary Data 1. Furthermore, some more details regarding the amplicon sequencing approaches have been provided in the Materials and Methods (line 738-747).

Major concern 3: Benefits of PE nucleases are unclear. – Resolved.

1. The authors have convinced me that this is an interesting activity and that this approach may be useful for applications in which total frequency of desired edits is the primary goal.

Major concern 4: Side effects of i53 are not properly monitored. – Resolved.

a. The authors conclusively address this.

[Authors' responses]

We thank the reviewer very much for commenting positively on the rest of our revision efforts.

Reviewer #2 (Remarks to the Author):

All issues have been sufficiently addressed.

Suggestion: please consider moving supplementary figures 11-14 to the main figures.

[Authors' responses]

We thank the reviewer for this suggestion.

Indeed, the information provided in Supplementary Fig. 11 was similar to those in Fig. 4B, D, with the latter dedicated to presenting the normalized editing levels by uPEn over PE5max (in HEK293T and U2OS cells, respectively). This part of the data presentation is maintained.

We have since included a graph to present the normalized levels of base conversions by uPEn over PEEn in HeLa cells in the current Fig. 4E. Such a graph has provided an accessible summary of the results related to those in Supplementary Fig. 12.

The results in Supplementary Fig. 13 mainly present experimental controls to exclude confounding factors for determining the relative performances of uPEn. To ensure an overall easy comprehension of our work by the readers, we elected to leave these results in the Supplementary Information.

We moved the data originally in Supplementary Fig. 14A-D to Fig. 5. These results benchmark the performances by uPEn for installing various types of small-block edits against the canonical PE platforms. The remaining Supplementary Fig. 14A demonstrate the corresponding summary of results (originally in Supplementary Fig. 14E).

- 1 Chen, P. J. *et al.* Enhanced prime editing systems by manipulating cellular determinants of editing outcomes. *Cell* **184**, 5635–5652 e5629, doi:10.1016/j.cell.2021.09.018 (2021).
- 2 Deriano, L. & Roth, D. B. Modernizing the nonhomologous end-joining repertoire: alternative and classical NHEJ share the stage. *Annu Rev Genet* **47**, 433–455, doi:10.1146/annurev-genet-110711-155540 (2013).